# Two Sides of the Coin: Ezrin/Radixin/Moesin and Merlin Control Membrane Structure and Contact Inhibition

**DOI:** 10.3390/ijms20081996

**Published:** 2019-04-23

**Authors:** Katharine A. Michie, Adam Bermeister, Neil O. Robertson, Sophia C. Goodchild, Paul M. G. Curmi

**Affiliations:** 1School of Physics, University of New South Wales, Sydney 2052, Australia; k.michie@unsw.edu.au (K.A.M.); adam.bermeister@gmail.com (A.B.); robertn1@mskcc.org (N.O.R.); 2Department of Molecular Sciences, Macquarie University, Sydney 2109, Australia; sophia.goodchild@mq.edu.au

**Keywords:** merlin, ezrin, moesin, radixin, FERM domain

## Abstract

The merlin-ERM (ezrin, radixin, moesin) family of proteins plays a central role in linking the cellular membranes to the cortical actin cytoskeleton. Merlin regulates contact inhibition and is an integral part of cell–cell junctions, while ERM proteins, ezrin, radixin and moesin, assist in the formation and maintenance of specialized plasma membrane structures and membrane vesicle structures. These two protein families share a common evolutionary history, having arisen and separated via gene duplication near the origin of metazoa. During approximately 0.5 billion years of evolution, the merlin and ERM family proteins have maintained both sequence and structural conservation to an extraordinary level. Comparing crystal structures of merlin-ERM proteins and their complexes, a picture emerges of the merlin-ERM proteins acting as switchable interaction hubs, assembling protein complexes on cellular membranes and linking them to the actin cytoskeleton. Given the high level of structural conservation between the merlin and ERM family proteins we speculate that they may function together.

## 1. Introduction

The paralogous proteins merlin (also called neurofibromatosis type 2 tumor suppressor, NF2 or schwannomin) and the ERMs—a group of proteins comprised of ezrin (also called cytovillin, villin-2 and p81), radixin and moesin (an acronym for membrane-organizing extension spike protein) provide an interesting platform on which to study the evolution, divergence and acquisition of functional differences in a family of highly similar and highly conserved proteins. The most well studied ERM, ezrin, has been implicated in the formation of specialized actin-linked plasma membrane structures. Merlin is also required for the formation of many membrane structures; however, it also interacts with various signaling pathways not just proximal to the membrane but in the cytoplasm and nucleus as well. As these proteins have nearly identical tertiary structures, subtle differences in how each merlin-ERM member interacts with other proteins and how they mediate control over numerous pathways seems to be the driver behind their functional differentiation.

The purpose of this review is to relate the cellular and biochemical data to the structure of these proteins in order to better understand their functional similarities and differences. Our perspective is that of structural biologists who follow the evolution of the merlin-ERM family from its emergence, which is synchronous to the appearance of metazoa.

## 2. Overview

Merlin-ERMs are multidomain proteins (Figure 1a) comprised of three defined parts—the N-terminal FERM (band 4.1 protein, ezrin, radixin, moesin) domain (also called the N-terminal ERM association domain or N-ERMAD), a central helical domain that is dominated by an extended, antiparallel coiled-coil (CC) and a short, mainly helical C-terminal domain (CTD, also referred to as the C-terminal ERM association domain or C-ERMAD) [1,2,3]. The current view is that the FERM domain binds to membranes, integral membrane proteins and scaffolding/effector proteins, while the C-terminal domain binds to the cortical actin cytoskeleton [4,5,6]. The merlin-ERMs act as linkers between membranes and the underlying actin cytoskeleton, and hence they can function in membrane remodeling, maintaining specialized membrane structures and vesicle trafficking [7,8].

The ERM proteins appear to have two states: an autoinhibited “closed” state and an active “open” form (Figure 1b,c, respectively; see Section 4.6). In the closed form, the CTD binds to a site on the FERM domain, presumably masking important interaction sites on both domains. Activation of ERM proteins is associated with membrane binding and phosphorylation, as will be discussed in this review.

Merlin is distinct from the ERMs with regards to activation and the open-closed dichotomy [9,10]. The function of merlin differs from the ERMs, in that it goes beyond linking membranes to the actin cytoskeleton and assembling protein complexes at the membrane. Merlin functions both at the membrane and in the nucleus, while ERMs are mainly restricted to membranes and the cytoplasm [2]. Merlin regulates cell proliferation and mediates contact inhibition (with the combination of the two contributing to its classification as a tumor suppressor) by participating in multiple signaling pathways, most importantly in Hippo signaling (also known as Salvador/Warts/Hippo pathway) and Wnt/β-catenin signaling [11]. In Hippo signaling, merlin forms a heterotrimer with two other Hippo components (WWC1 and FRMD6, another FERM domain containing protein) to activate the core kinase in response to extracellular stimuli. This results in the activation of transcription factors important for cell proliferation and apoptosis [12]. Merlin’s role in Wnt/β-catenin signaling, a major pathway involving adherens junctions, is less clear, however there is strong evidence of changes in merlin’s expression level directly affecting β-catenin’s ability to remain in the nucleus and act as a transcription factor for genes controlling contact inhibition [13].

## 3. Evolution and Biology of Merlin-ERM (Ezrin, Radixin and Moesin) Proteins

### 3.1. Evolution of the Merlin-ERM Family of Protein

In this Section, we have probed the evolution of the merlin-ERM protein family. Using the program BLAST [14], we searched the National Center for Biotechnology Information (NCBI, US National Library of Medicine) non-redundant protein sequence database for distal members of the merlin-ERM family. Using multiple sequence alignment tools (MUSCLE [15]), we generated alignments and neighbor-joining phylogenetic trees to survey merlin-ERM evolution.

Merlin-ERM proteins appear to have evolved near the emergence of metazoa [16]. The only non-metazoan organisms carrying merlin-ERM genes are the choanoflagellates, the closest single-celled eukaryotes to the metazoa, where both metazoa and choanoflagellates belong to the clade opisthokonta. The other members of the opisthokonta, including fungi, do not appear to contain merlin-ERM genes.

Separate merlin and ERM genes appear to have evolved early [16] with a clear partitioning of merlin versus ERM genes in phylogenetic trees (Figure 2). There is evidence that choanoflagellates have separate ERM and merlin genes, in particular, *Salpingoeca rosetta* (Figure 2). Pairs of merlin and ERM genes are also present in early branching metazoa such as porifera (*Amphimedon queenslandica*, Great Barrier Reef sponge) and cnidaria (*Nematostella vectensis*, the starlet sea anemone). Most bilateria have both merlin and ERM genes, apart from platyhelminthes (flatworms), which appear to have a single merlin-ERM that is difficult to categorize as either merlin or ERM (Figure 2). These observations suggest that the emergence of merlin-ERM genes and their divergence into separate merlin and ERM proteins are linked to the evolution of cell differentiation, specialization and multicellular organization in metazoa, with choanoflagellates such as *S. rosetta* showing a primitive level of cell differentiation and specialization.

The evolution of vertebrates is accompanied by the generation of at least three ERM paralogs, namely ezrin, radixin and moesin, while merlin maintains a single gene. The three vertebrate ERMs may be the outcome of two rounds of whole genome duplication as per the Ohno hypothesis [17]. Several species of fish have also undergone an additional round of whole genome duplication, resulting in 6–8 ERMs plus two copies of merlin [18]. In contrast, invertebrates have only 2 genes encoding merlin-ERM proteins—one gene encodes a merlin homolog and the other is a general ERM protein sharing traits with all the ERM proteins (commonly called moesin in the literature although this classification is somewhat arbitrary).

Multiple sequence alignments of human merlin-ERM (Figure 3) show high conservation over the merlin-ERM family (37% identity) with a higher degree of sequence identity over the ERM proteins (67%), in keeping with their later divergence at the chordate–vertebrate boundary (Table 1). The FERM domains show a higher level of sequence identity compared with the central helical domain and the CTD.

### 3.2. Isoforms, Biological Expression and Disease

Both ezrin and moesin appear to be expressed as a single isoform from the genes *EZR* (also called *VIL2*) and *MSN*, respectively, and are found in most tissue types. Moesin is found mainly in the filopodia and is found in the highest levels in appendix, bone marrow, lung, lymph node, spleen and thyroid. A single missense mutation (R171W) has been identified as the cause of an X-linked primary immunodeficiency [22]. Consistent with a role in immunodeficiency, there are a number of reports linking moesin to roles in viral infection including measles [23], HIV [24] and hepatitis C [25]. Moesin has also been observed to have raised levels in muscular dystrophies [26], and cervical, breast and lung cancers [27,28,29].

Unlike moesin, there is no clinical data revealing a particular disease or disorder directly attributed to errors in ezrin. While a number of missense mutations have been identified in patient samples, no phenotypic effects have been discerned. Currently ezrin’s biological role has been examined by specific cell biology experiments and its expression and localization have been monitored and reported to be aberrant in a number of cancers (including pediatric osteocarcinoma and rhabdomyosarcoma, and adult breast, lung, colorectal and pancreatic amongst others, in particular to the process of metastasis [30,31,32,33,34,35,36,37,38]) and during viral infection [24,25,39,40,41]. Ezrin is expressed in the gastrointestinal tract, epithelial cells, kidney, spleen and lymph nodes [42].

In contrast to ezrin and moesin, both merlin and radixin are expressed in vivo as various isoforms (see www.proteinatlas.org [43]). Why this is the case is still to be resolved.

Merlin is comparatively complex when compared to ezrin and moesin. Due to this complexity there is a great deal of confusion in the consistency of the naming of its isoforms. There are 10 different isoforms of merlin as reported by Uniprot (P35240), formed by alternative splicing of the 17 exons of the *NF2* gene. Isoform 1 (also known as I) is the canonical sequence and results from the expression of exons 1–15 along with exon 17 and it bears the closest sequence relationship to the ERM proteins. Merlin isoforms 1 and 2 are the predominantly expressed isoforms in human tissue, whilst the other 8 are seldom expressed [44].

One key difference between merlin isoform 1 and the remaining isoforms is the nature of the merlin C-terminus from Leu580 onwards (see Figure 3 for sequence numbering). Whilst isoform 1 has a typical ERM C-terminus, isoforms 2 (also known as II and annotated as ‘isoform 3′ on Uniprot —note: Uniprot ‘isoform 2′ is probably spurious, resulting from a frame shift error in the particular assembly), 4, 5, 6 and 8 replace the C-terminal sequence by including exon 16 which contains an in frame stop codon, thus, altering the properties of the C-terminus. The new C-terminus is 5 residues shorter, more hydrophilic at the very C-terminus but more hydrophobic closer to the expressed splice junction as well as being potentially non-helical [44]. These properties distinguish this C-terminus from that of the major merlin isoform 1 and the ERM proteins. The structure of this altered C-terminus is most likely to differ from merlin isoform 1 and the ERM proteins; however, it remains to be resolved whether this altered C-terminus weakens the FERM–CTD interaction in these merlin isoforms. From an evolutionary perspective, the alternative C-terminus seen in merlin isoforms encoding exon 16 has only been observed in the clade sarcopterygii and all descendants including all tetrapods (including mammals) and coelacanths (lobe-finned fish, *Latimeria chalumnae*), but excluding ray-finned fish (the majority of species of extant bony fish). This suggests that this merlin variant arose early in vertebrate evolution around the time that vertebrates invaded dry land.

The remaining merlin isoforms, isoforms 7 and 10, are also missing large portions of the C-terminus, thus, alteration of the merlin C-terminus appears to be a common feature among the merlin isoforms.

Apart from the alternative C-terminus, merlin isoforms 4, 5, and 6 are missing almost the whole F1 lobe of the FERM domain (residues 39–121 for isoform 4; 81–121 for isoform 5; and 39–80 for isoform 6; see Figure 3 for human merlin sequence), suggesting that in these isoforms, the FERM structure is likely to be altered. These three isoforms are expressed moderately. Isoform 8 is expressed at low frequency. It also carries the C-terminal sequence variation as per isoforms 3–6 as well as a deletion at the beginning of the helical domain (residues 335–363), which includes the C-terminus of helix α1H and part of the N-terminus of helix α2H prior to the coiled-coil region. Isoforms 7 and 9 are not found in adult tissues. Isoform 10 is found in adult human retina. These last three merlin isoforms are significantly truncated, encoding a protein half the length of the full sized merlin. Isoform 7 encodes only part of the FERM domain (residues 1–259 where the C-terminal half of the F3 lobe is missing). Isoform 9 encodes the FERM F1 lobe and half of the FERM F2 lobe with a small section of the CTD C-terminus fused on. Isoform 10 is a confusing fusion of two β strands of the FERM F1 lobe and one and a half helices of the F2 lobe.

It has been suggested that only merlin isoform 1 can suppress tumors [45] although data are conflicting. A large number of mutations within merlin are known to cause or contribute to the diseases neurofibromatosis 2 (NF2) [46], schwannomatosis 1 (SWNTS1) [47] and a form of malignant mesothelioma (MESOM) [48].

Radixin (expressed from *RDX*) is similar to merlin in that it is expressed as at least 5 isoforms by alternative splicing. It is found in high levels in the adrenal glands and kidney. The main isoform, isoform 1, is a 583 amino acid protein comprising a complete merlin-ERM protein that shares high sequence identity with the other ERMs (sequence shown in Figure 3). The most common feature of the remaining isoforms is a 21 amino acid extension on the C-terminus, which is seen in isoforms 2, 3 and 5. Isoform 5 is identical to isoform 1 apart from this extension. Isoform 2 is missing the complete FERM domain (residues 1–347) with the 21 amino acid extension following the C-terminus (equivalent to residue 583). Isoform 3 is comprised of a part of the F1 and F2 lobes of the FERM and the same 21 amino acid C-terminal extension as isoform 2, with a complete length of 200 amino acids. Isoform 4 has a short 20 residue peptide that completely replaces the F1 and F2 lobes of the FERM domain. It has a native radixin C-terminus without the 21 residue extension.

The precise molecular function of radixin is largely a mystery, similar to ezrin and moesin, however, it is known to be present in the hair cell stereocilia of the inner ear [49,50,51,52,53]. Clues have been garnered from clinical data where it has been observed that a single point mutation in radixin (D578N) gives rise to a form of non-syndromic sensorineural autosomal recessive deafness (DFNB24) in humans [53]. This residue is on the terminal helix of the CTD (helix α4C, see Figure 3 and Figure 4) of isoform 1 (normal merlin-ERM fold) and points away from the FERM domain in the closed form. It is not obvious why such a mutation should give rise to this phenotype. In radixin knockout mice the two main phenotypes are hearing impairment and hyperbilirubinemia, although this defective liver phenotype has not been observed in human patients with mutations in radixin. Similar to ezrin and moesin, an increased level of radixin expression has been observed in many tumor tissues [54] and it has also been implicated in the viral infection process for several viruses [25].

It is interesting to think about why merlin and radixin are expressed as so many different isoforms, typically with C-terminal alterations, yet ezrin and moesin appear to be expressed as single isoforms. Analysis of merlin isoforms shows that many of them are minimally expressed. Some of the isoforms are comprised of only small sections of the protein, certainly not the complete architectural unit comprising the FERM, the helical domain and the CTD. For merlin, with a number of roles in signaling, it is possible that these short isoforms are carrying out purely signaling roles. For radixin, it is less obvious why there are so many isoforms.

## 4. Merlin-ERM Structure

As per Section 2, each merlin-ERM can be divided into three domains: the N-terminal FERM domain (human ERM residues 1–296; human merlin 1–312); the central helical domain (ERM residues 297–469; merlin 313–478) and the CTD (ezrin 470–586; radixin 470–583; moesin 470–577; merlin 479–595). Figure 3 shows these domains mapped onto a sequence alignment of human merlin-ERM.

Only one full-length ERM protein structure (PDB 2I1K, from the moth *Spodoptera frugiperda*) has been solved in the closed, autoinhibited state (henceforth referred to as *Sf*moesin Figure 4a) [55]. This protein represents a common ancestral ERM protein monomer prior to divergence into the separate ezrin, radixin and moesin paralogs. Wild-type Sfmoesin was directly purified from *S. frugiperda* ovarian cells, crystallized and its structure determined using standard methodologies [55]. Comparison of this full-length *Sf*moesin structure with higher-resolution structures of mammalian FERM domains and FERM:CTD complexes, suggests that this full-length structure represents the ERM family well. Structure determination of the full-length proteins has been difficult due to the propensity of the central helical domain to undergo partial proteolytic cleavage [56]. The full-length *Sf*moesin structure reveals that the major portion of the helical domain forms an elongated antiparallel coiled-coil extending from the globular FERM:CTD complex (α helices: α2H and α3H; Figure 3 and Figure 4a). The exposed, protruding nature of this coiled-coil structure may explain why proteolytic cleavage is often observed within the central helical region and the polypeptides connecting it to the FERM and CTD.

There are extensive structural data for isolated domains from the merlin-ERM proteins. Crystal structures have been solved for the FERM domain for all 4 of the proteins representing the merlin-ERM proteins [56,57,58,59,60] (Figure 4c). In each case, isolated FERM domains were heterologously expressed in *Escherichia coli* and the protein purified, crystallized and the structure determined via standard methods. Furthermore, crystal structures of the complex between the FERM domain and the CTD (without the central helical domain) from merlin, ezrin and moesin have also been determined [56,61,62] (Figure 4d). As expected from the high degree of sequence identity (Figure 3 and Table 1), the structures are near identical (Figure 4c,d). We note that for the structures of the merlin [62], and moesin [61] FERM:CTD complexes, the FERM and CTD domains were expressed separately in *E. coli* and the complex formed in vitro. In contrast, the crystal structures of the ezrin FERM:CTD complex were obtained from the crystallization of full-length ezrin where inadvertent limited proteolysis of the protein occurred during crystallization [56].

### 4.1. The FERM (Band 4.1 Protein, Ezrin, Radixin, Moesin) Domain

The FERM domain is an evolutionarily conserved 30 kDa composite domain [64] made up of three subdomains called F1, F2 and F3: a ubiquitin-like domain; a four-helix bundle; and a phosphotyrosine-binding/peckstrin homology (PH) domain, respectively (see Figure 3 and Figure 4). Together, these subdomains form a tri-lobed, cloverleaf structure (Figure 4c), roughly 70 Å by 70 Å across and 40 Å deep [58,61]. The tri-lobed structure is common to almost all FERM-containing proteins. An important exception to this is the integrin-cytoskeleton linker talin [65]. In talin, the F2 and F3 subdomains interact as per all other cloverleaf FERM domains, however, subdomain F1 is displaced, resulting in a linear subdomain arrangement. Talin also includes a second ubiquitin-like subdomain, F0, at the N-terminus, which extends the linear structure [65]. We note that the properties of F1 plus the F1-F2 linker and helix α1F2 in F2 are such that talin could potentially form a cloverleaf arrangement as per the other FERM domains. Thus, it is possible that talin can switch between the linear FERM and the cloverleaf FERM under appropriate circumstances.

The triangular arrangement of subdomains within the FERM cloverleaf gives rise to inter subdomain clefts between F1, F2 and F3 (Figure 5a). Important interactions between the subdomains are conserved within these clefts and these stabilize the whole FERM domain. The interactions involve hydrophobic and polar contacts including hydrogen bonds and salt bridges. One of the three subdomain clefts has a positively charged/basic surface (between F1 and F3—labeled ‘a’ Figure 5a,) while the cleft between F2 and F3 has a negatively charged/acidic surface (labeled ‘b’ in Figure 5a) and, finally, the remaining cleft (between F1 and F2) is more open than the other two (labeled ‘c’ in Figure 5a) [58]. These general features are conserved across the merlin-ERM proteins.

On one side of the FERM domain, under the apex of the three lobes, is a deep hydrophobic pocket (Figure 5a, right panel). Although there is no evidence that this pocket is utilized by merlin-ERM proteins, the corresponding pocket is used to bind cargo by the myosin MyTH4-FERM domain [67,68,69]. The opposite face presents a solvent exposed acidic/negatively (red) charged patch (Figure 5a, center). The C-terminal domain (discussed in Section 4.3) is able to make strong interactions with the FERM domain, binding in an extended single-layer structure across the negatively charged surface of the F2 and F3 subdomains (Figure 5, center in red). When the CTD is ‘peeled away’ from the FERM domain, calculations of electrostatic surface potentials show that the two exposed surfaces contain complementary charges (Figure 5b).

There are two crystal structures of the merlin-ERM FERM domain bound to phosphatidylinositol 4,5-bisphosphate (PI(4,5)P_2_) mimetics, either IP_3_ or short chain PI(4,5)P_2_ (see Section 7.1 below, plus Figure 10f radixin PDB 1GC6 [58] and Figure 10g merlin PDB 6CDS [70], respectively). The two structures show the PI(4,5)P_2_ mimetic binding in the F1/F3 cleft (Figure 5a, cleft a), albeit in different locations within the cleft (see Section 7.1). We note that the precise binding site is different in the two structures and the electron density for the PI(4,5)P_2_ mimetic is weak in both cases.

The FERM domain is a defining feature of a large number of proteins whose biological roles include linking the cytoskeleton to the plasma membrane. These proteins include: PTP-BAS (a protein tyrosine phosphatase that associates with the cytoplasmic tail of the cell-surface receptor Fas); the erythroid protein Band 4.1; CORACLE; CDEP; and the FERM domain at the C-terminus of Myosin VII, X, XII and XV [64,71]. The role of FERM in all these proteins is still to be determined, however, as discussed in Section 7.1, this domain plays an important, direct role in the merlin-ERM proteins’ ability to bind to membranes.

### 4.2. The Central Helical Domain

The central helical domain lies between the FERM domain and the CTD (Figure 1, Figure 3 and Figure 4). It is composed of three helices: α1H, α2H and α3H, where the latter two combine to form an elongated antiparallel coiled-coil protruding from the globular FERM:CTD complex [55] (Figure 4a). Helix α1H lies along FERM subdomain F2 and it is essentially part of the globular FERM domain. This can be seen in the extended radixin FERM constructs used to determine the crystal structures of radixin:ICAM-2 [72], radixin:CD43 [73] and radixin:PSGL-1 [74]. In these structures, helix α1H adopts the same packing with respect to the FERM domain as seen in the structure of the full-length *Sf*moesin [55]. Helix α1H can also be seen in the same site in the structure of the extended moesin FERM domain (residues 1–346), although in this case, α1H extends to the end of the construct where the extension appears to be supported by crystal contacts [59].

An unusual feature of the merlin-ERM monomer structure is the coiled-coil that protrudes from the globular FERM:CTD structure. This coiled-coil is formed by the seven C-terminal heptads from helix α2H and all of helix α3H of the central helical domain. The coiled-coil is maintained by a classical heptad repeat comprising hydrophobic residues at the ‘a’ and ‘d’ positions. We note that the N-terminal region of helix α2H also maintains a heptad repeat (two heptads) although it packs against the body of the FERM domain (subdomains F1 and F2; see Figure 4a).

The length of the coiled-coil appears to be conserved for both merlin and the ERM proteins throughout metazoan evolution suggesting that the length of this coiled-coil structure has biological relevance [56]. For merlin, the length of the coiled-coil has not changed during metazoan evolution. For the ERM proteins, there was an insertion of a complete heptad at the N-terminus of helix α2H that occurred at the chordate–vertebrate boundary (Figure 3) [55,56], however, this region precedes the coiled-coil.

As there are no high-resolution structures of intact vertebrate ERM proteins, it is unclear how the seven residue insertion alters the arrangement of the coiled-coil protrusion with respect to the globular FERM:CTD complex. The length of the coiled-coil is likely to remain the same as the insertion precedes the coiled-coil with no corresponding change to helix α3H and the central hinge (see below) between helices α2H and α3H is unchanged. Understanding the consequences of the insertion in the vertebrate ERMs will have to await the determination of a complete, high-resolution structure.

#### The Hinge Sequence

Sequence analysis (for example COILS [75]) shows that there is a clear hinge between α2H and α3H in the coiled-coil region for all merlin-ERM proteins (Figure 3, region between αH2 and αH3). Within the hinge, the amino acid sequence shows a remarked reduction in its predicted ability to form a coiled-coil (i.e., the a and d heptad repeat breaks down). The hinge corresponds precisely to the loop connecting helices α2H and α3H in the crystal structure of the *Sf*moesin monomer [55].

The length of the hinge sequence is strictly conserved in all known merlin-ERM proteins. More importantly, the hinge sequence preserves the register of the heptad repeat between helices α 2H and α3H of the coiled-coil, allowing the formation of an extended, continuous α helix across the whole length of this region. It has been proposed that such an extended α helix is the basis of the structure of the ezrin domain swapped dimer that has been determined at low resolution by SAXS (Figure 4b) [56]. The conservation of the hinge in all merlin-ERM proteins may explain the formation of homo- and heterodimers (see Section 4.5).

### 4.3. The C-Terminal Domain (CTD) or C-Terminal ERM Association Domain (C-ERMAD)

Merlin-ERM proteins contain a C-terminal domain (CTD) that is much shorter and more divergent in sequence than their corresponding N-terminal FERM domains when compared across the family. There are significant differences between merlin and the ERMs in this region (see Section 8.2 and Section 8.3).

Two main points to note are that, firstly, the CTD of merlin-ERM carries out important regulatory roles by interacting with the FERM domain. This interaction masks and unmasks critical sites for protein–protein interactions (see Section 5) and probably, protein–lipid interactions on the FERM and protein–protein interactions within the CTD. Secondly, the CTD contains strictly conserved residues that have been shown to be post-translationally modified (see Section 6).

The only atomic structural information regarding the CTD comes from crystal structures of full-length *Sf*moesin [55], and the FERM:CTD complexes of human moesin [61], human ezrin [56] and human merlin [62]. In these complexes, the CTD forms a flat, essentially two-dimensional structure that forms an outer coat on the FERM domain (Figure 4a). The structure of the CTD can be broken into two segments. On leaving the central coiled-coil, the CTD adopts a hairpin structure comprising the proline-rich/low complexity segment followed by a β-strand, extending the outer β-sheet of FERM subdomain F3 (Figure 3 and Figure 4a). This hairpin lies in the groove between FERM subdomains F1 and F3, thus it may obstruct the PI(4,5)P_2_ binding site (see below, Section 7.1 and associated Figure 10 for details). The C-terminal part of the CTD forms a flat, α helical structure that covers FERM subdomains F2 and F3 (Figure 4a,d). There are four extended α helices in this region (α1C, α2C, α3C and α4C, Figure 3 and Figure 4d).

Sequence analysis (and conservation) indicates that the merlin CTD is likely to adopt the same structure as the ERM CTD when complexed to the FERM domain. This is largely borne out by the only structure of a merlin FERM:CTD complex; however, this structure deviates from this expectation in that the segment between where the β strand is expected to end (absent in the crystal structure) and the first extended α helix of the CTD (α1C) form one long α helix (Thr512–Leu546, see Figure 3 for sequence) [62]. This merlin structure contains a phosphomimetic mutation S518D, (where Ser518 is unique to merlin). The mutant Asp518 lies near the N-terminus of the observed extended CTD helix. Phosphorylation of merlin Ser518 does not alter the conformation/stability of the FERM:CTD complex, however, it alters binding of merlin to angiomotin and hence cellular signaling [62]. We note that this long α helix observed in the merlin FERM:CTD complex is supported by crystal-packing interactions.

There is no evidence that the isolated CTD forms an independently folded structure. By observing the CTD when complexed to the FERM, it is likely that its structure is partially dictated by the protein to which it is bound. It is probable that the CTD may adopt different tertiary structures when bound to other partners (i.e., it is thought that ezrin interacts with actin via the CTD and it is possible that the CTD will form a somewhat different structure in this case).

#### Polyproline Region

At the end of the coiled-coil region is a polyproline motif in both (vertebrate) radixin and ezrin (see Figure 3) [76]. Similarly there is a proline-rich region in mammalian merlins (see Figure 6 for the primary sequence alignment of this region across the human merlin-ERM family plus *Sf*moesin). This feature is surprisingly absent in vertebrate moesin [77]. The role of the prolines is not understood, however it is known that ribosomes can stall at consecutive prolines during translation [78,79] and thus such proline-rich stretches may be important for domain folding. In the case of merlin-ERM it seems this is unlikely as the proline-rich motif is at the end of the coiled-coil (α helical) region that is not expected to need a pause to aid folding.

A more interesting possibility arises from the knowledge that polyproline regions often form left-handed polyproline type II helices (PPII helices). These structural motifs are commonly used as interaction sites that bind proteins, DNA, as well as membranes [81,82,83,84,85,86,87]. The PPII helix arises when all residues adopt backbone dihedral angles (ϕ and ψ) of approximately −75° and 150°, and all peptide bonds are *trans*. The helix is relatively open and extended, containing no internal hydrogen bonds. Other amino acids are able to adopt this conformation (up to 46% of polyproline helices in folded protein structures have no prolines [88]) and may be present in these motifs with the most common amino acids in this secondary structure type being alanine and leucine, followed by arginine, methionine and lysine [89].

Currently, there is no direct structural data on the polyproline tract/proline-rich region for vertebrate ezrin/radixin or mammalian merlin. Insect ERMs do not have a proline-rich domain, instead, they often have a poly-histidine tract followed by a low-complexity negatively charged region (see *S. frugiperda* sequence in Figure 6). The crystal structure of full-length *Sf*moesin shows that the poly-histidine segment (^462^PQHHH^466^) forms a single turn 3_10_-helix (Figure 4a, single turn orange ribbon above subdomain F1, blue), while the following segment, ^467^VAERAD^472^ forms a left-handed PPII helix (Figure 4a, orange-red string protruding in the cleft between F1, blue, and F3, green) [55]. This PPII helix lies in the groove between FERM subdomains F1 and F3, running anti-parallel to the subdomain F3 α helix (Figure 4a). There is a break in the electron density after Asp472, with the structure resuming with Glu486, which starts a β strand packing against the outer β sheet in subdomain F3 (Figure 4a, orange arrow against green arrow in F3). Thus, based on the *Sf*moesin crystal structure, it is quite possible that vertebrate ezrin/radixin and mammalian merlins contain a left-handed PPII helix just after the coiled-coil domain.

PPII helices have been identified to be important for interactions with a number of ligands, in particular the SH3 regulatory domain of numerous kinases interact with their substrates via a polyproline motif (specifically PPII helices) [81,82,83,84,85,86,87]. It is possible that the polyproline domain is an interaction motif for an activating kinase. The location of the putative PPII helix in the merlin-ERMs may indicate that it is involved in protein activation. In order to bind to a partner protein, the PPII would leave the slot between FERM subdomains F1 and F3. This is likely to release the additional CTD β strand from the outer β sheet of subdomain F3. The net result may be to facilitate membrane binding, particularly to PI(4,5)P_2_ (see Section 7.1).

### 4.4. Interaction between the CTD and FERM Domain and Implications

Merlin-ERM proteins all share a conserved potential to form a FERM:CTD complex enabling the regulation of the conformational and oligomeric states of the proteins (Figure 4a, the CTD is shown in red). The interaction between the FERM and the CTD is mediated primarily by the burial of conserved hydrophobic side chains from helices within the CTD into conserved hydrophobic pockets of the F2 and F3 subdomains [61,62], although there are notable electrostatic interactions (Figure 5b shows the electrostatic potential of the interacting surfaces from FERM and CTD domains). The CTD has few intra-chain interactions, suggesting most of the structure forms upon binding to the FERM domain [61]. Binding of the CTD to the FERM causes small movements in parts of the F2 and F3 subdomains of the FERM domain, allowing for accommodation of α-helices from the CTD [56,58,59].

What is striking about the FERM–CTD interaction is that the individual interactions are conserved across the merlin-ERM family. Helix α1C has a conserved hydrophobic face that binds to a hydrophobic region on FERM subdomain F2 (Figure 5b the binding surface of α1C is colored white indicating its non-polar nature). The N-terminus of helix α2C makes conserved hydrophobic interactions with the interface between FERM subdomains F2 and F3 (Figure 5b α2C is also colored white). More striking is a conserved polar face of helix α2C that forms three conserved side chain-side chain hydrogen bonds with conserved residues in FERM subdomains F2 and F3 (ezrin: Asp551–Tyr116, His554–Glu199 and Asn557–Thr235; merlin: Asp559–Tyr132, His562–Glu215 but Asn565 is disordered in the crystal structure although Thr251 is observed).

The interaction between helix α3C and FERM subdomains F2 and F3 is more indirect. There is a water-filled cavity separating helix α3C from the FERM, with the exception of a conserved hydrophobic interaction at the C-terminus of this helix. Helix α3C comes into close proximity with the portion of the CTD connecting the β-strand to helix α1C. These two segments are highly positively charged in all merlin-ERMs with multiple conserved lysines and arginines. These segments lie above a conserved acidic surface of the FERM domain (Figure 5a, central panel). Thus, electrostatics is expected to play a role in stabilizing the interaction between FERM and this portion of the CTD. ‘Peeling away’ the CTD from the FERM domain in the ezrin FERM:CTD crystal structure (Figure 5b, left panel) and calculating the electrostatic potential of the exposed surfaces shows strong electrostatic complementarity (Figure 5b, right panel). Finally, we note that ERM Thr567, whose phosphorylation is associated with ERM activation, is located in helix α3C. It lies adjacent to FERM subdomain F3 in the FERM:CTD complex.

Helix α4C appears to lock the CTD to the FERM domain. There is a conserved hydrophobic face that interacts with FERM subdomain F3. In particular, there is a conserved phenylalanine (ezrin—Phe583; merlin—Phe592) that slots into the hydrophobic core of subdomain F3. This motif is used by ERM binding proteins (see Section 5, below). Additionally, there is a conserved salt bridge between ezrin Arg579 (merlin Arg588) and FERM ezrin Glu244 (merlin Glu260) anchoring helix α4C to subdomain F3. The contact between these residues’ side chains is approximately 2.5 Å (taken from crystal structures) implying that it is relatively strong.

It is noteworthy that in merlin isoform 2 (also known as isoform II, see Section 3.2), the result of alternative splicing alters the sequence from the C-terminus of helix α3C. The new sequence, ^580^PQAQGRRPICI^590^, does not appear to code for helix α4C, hence, it is not clear that merlin isoform 2 is capable of forming a FERM:CTD complex, distinguishing this isoform from the majority of merlin-ERM proteins. We also note that the minor isoforms of merlin (isoforms 3–10) also lack the C-terminus of the CTD and hence may not form a FERM:CTD complex.

### 4.5. Oligomeric States of Merlin-ERM Proteins

The formation of an intra-molecular ‘head-to-tail’ interaction between the FERM and CTD domains has several implications, including the possibility of forming multiple oligomerization states. Indeed, Förster Resonance Energy Transfer (FRET) and co-immunoprecipitation experiments have shown that ezrin oligomers form in cells [77,90], and dimeric ezrin has been purified from placental tissue [91,92]. The interaction of the FERM and CTD is strong with little exchange between monomers and dimers occurring [92,93,94,95]. Small-angle X-ray scattering (SAXS) studies of purified ezrin dimer have shown that it largely adopts a domain-swapped dumbbell structure (Figure 4b) [56].

Recent cryo-electron microscopy studies have shown that both WT ezrin and the phosphomimetic mutant, T567D, can juxtapose PI(4,5)P_2_ containing membranes [96]. The region between the two adjacent membranes contains a dense, brush-like structure comprising ezrin molecules. The separation between the two membranes is consistent with the dimensions of the domain-swapped ezrin dimer [56] as the structural unit linking the membranes; however, these authors observe a slight contraction in the ezrin brush for the phosphomimetic mutant (separation between centers of membrane-bound globular domains: T567D 24.1 ± 1.3 nm versus WT 28.7 ± 1.2 nm) [96]. The dense, brush-like structure implies that ezrin dimers are self-associating laterally on the membrane surface.

With the high level of conservation across the FERM and CTDs of the merlin-ERM family, it is not surprising that interactions between different members (hetero-associations) of the merlin-ERM family have also been reported. An interaction between merlin and ezrin, mediated by interactions between the CTD and FERM domains, has been detected in pull-down, co-immunoprecipitation and yeast-two hybrid experiments [93,97,98]. These experiments have shown that both the ezrin and merlin can interact, however the heterodimerization of merlin isoforms 1 and 2 (also known as I and II, respectively) is stronger than that between either merlin isoform and ezrin [98]. The ezrin CTD and merlin CTD have a higher affinity for the ezrin FERM than for the merlin FERM. Furthermore, ezrin FERM preferentially binds ezrin CTD over merlin CTD [93]. Interactions between merlin and moesin have also been shown using affinity co-electrophoresis (ACE) [99] and moesin and ezrin have been co-immunoprecipitated from the epidermal carcinoma cell line A431 [100].

Merlin and the ERM proteins are intimately structurally related, they are observed to co-localize in cells and have been shown to form heterodimers. These observations lead to the question: “Do merlin and the ERMs function directly together?” The differential affinities between the interacting domains of the merlin-ERM proteins hint at a mechanism for controlling heterodimerization. In vivo, overexpression of ezrin inactivates the tumor suppressor function of merlin in a glioblastoma cell line [101]. In this study, ezrin was observed to directly interact with merlin and alter its subcellular localization [101]. This study assessed some chimeric fusions of ezrin and merlin and also saw a reduction in the ezrin expression level as merlin over-expression was increased. These experiments indicate that merlin and ezrin possibly work together to control cell growth.

### 4.6. The Open and Closed Forms of ERM Proteins

One of the most studied structural states of the merlin-ERM family of proteins is commonly referred to as the ‘closed’ state, where the coiled-coil in the helical domain folds back along itself and the FERM and CTD interact forming discrete compact, globular structure (Figure 1b and Figure 4a) [55,56]. The FERM–CTD interaction is responsible for the formation of both the closed monomer and dimer states of ezrin (see Figure 1b and Figure 4a for the monomer and 1d and 4b for a model of the dimeric state). Recently, we have determined the crystal structures of the ezrin FERM:CTD complex from purified monomer and dimer fractions. The structures lack the central helical domain due to limited proteolysis during crystallization. However, both fractions result in almost identical FERM:CTD structures, suggesting the interaction takes place in both the closed monomer and closed dimer forms [56].

The merlin-ERM proteins are also thought to exist in an open conformation whereby the FERM and CTD do not interact (Figure 1c). It is unlikely that the coiled-coil region of the central helical domain is extended in this conformation, as a lone extended α helix would expose a large hydrophobic surface to the cytoplasmic medium (note: the merlin-ERM helical domain sequences are not compatible with a SAH domain—stable, single α helix [102]). Thus, such a structure is likely to be a short-lived intermediate state unless other cellular components can bind to the exposed α helices of the central helical domain and/or bind to the newly separated FERM and CTD, preventing their re-association into a closed form.

It is currently unknown whether the open form arises spontaneously with the FERM and CTD simply coming apart randomly or whether an active process is required for this structural transition to arise. For example, a conformational change in the FERM on binding lipid may cause the release of the CTD. The formation of an open form would only be initiated or stabilized by certain merlin–ERM interactions with other components that may sequester the FERM and/or CTD away from rebinding to each other or other merlin-ERM proteins, or that particular interactions may cause a conformational change that prohibits/destroys the FERM–CTD interaction. There is no high-resolution data describing an isolated, full-length open form, however, certain protein–protein interactions would only be able to occur when the interactions between the FERM and CTD are broken and, thus, the open form (either alone, chaperoned or stabilized by an intermediate binding partner) must occur in cells.

In addition to burying the actin-binding site, the closed monomer buries a number of residues known to be important for merlin-ERM protein–protein interactions (Section 5, Section 7 and Section 8 provide more information regarding protein-protein interactions with merlin-ERMs). It is currently thought that two factors play roles in the process of opening the molecule. 1: the binding of the FERM domain to PI(4,5)P_2_ (discussed in Section 7.1); and 2: the post-translational modification of a number of residues (see Section 6). For example, the phosphorylation of a conserved threonine (Thr567 in the CTD for ezrin [103], Thr558 for moesin and Thr564 of radixin—see Section 6.1) is thought to shift the equilibrium of ERM towards the open state.

## 5. Complexes between Merlin-ERM Protein FERM Domains and Their Binding Partners: The FERM as a Signaling Hub and Integrator

The FERM domains of the merlin-ERM proteins act like signaling hubs, binding numerous integral membrane and cytoplasmic proteins to various sites on the FERM domain. This is likely to integrate numerous cellular and extracellular signals to control membrane structure and cell function. In this Section, we review the structural data for the interactions between merlin-ERMs and their binding partners. We have not assessed whether interactions have been determined in vivo versus in vitro. To explore this, we refer the reader to the primary references cited in this Section.

The interaction between the merlin-ERM FERM and integral membrane proteins appears to be mediated by the intrinsically disordered cytoplasmic tails of these proteins. Merlin-ERM FERM binding partner integral membrane proteins include: CD44, CD43 (also known as leukosialin), ICAM-1, ICAM-3, PSGL-1, neprilysin and membrane type 1 matrix metalloproteinase (MT1-MMP). The crystal structures of peptides representing the cytoplasmic interaction domains of these integral membrane proteins with the radixin FERM domain reveal common binding modes. CD44 [104], CD43 [73], ICAM-2 [72], PSGL-1 [74] and neprilysin [105] peptides all bind to FERM subdomain F3 as an additional β strand, extending the outer β sheet of subdomain F3 by binding to β5F3 in the groove above F3 helix α1F3 (Figure 7, top row, central panel shows the radixin FERM:CD44 peptide complex as an example). The binding of the MT1-MMP is distinct, while it also binds by extending a β sheet, this time it binds to subdomain F1 by binding to β2F1 (Figure 7, top row, right hand panel) [106]. Finally, the merlin-ERMs are involved in establishing and maintaining epithelial cell polarity. The *Drosophila melanogaster* type I transmembrane protein crumbs (Crb) has a 37 residue cytoplasmic tail that contains a FERM binding motif. The crystal structure of the complex of this peptide with the mouse moesin FERM shows that the first part of the FERM binding region binds to the same site as the majority of cytoplasmic domains (i.e., as a β-strand extending the F3 β sheet by binding to β5F3), however, it doubles back, forming a β hairpin adding an additional β strand [107]. A second FERM binding region in the same peptide (at the very C-terminus of Crb) then binds along the groove between FERM subdomains F1 and F3 (Figure 7, top row, left hand panel).

The merlin-ERM FERM domain also binds soluble proteins, often recruiting them to the membrane-cytoplasm interface. The scaffolding protein EBP50, also known as NHERF1 (and the related NHERF2), binds to the FERM domain by displacing the final helix (α4C) of the CTD (Figure 7, bottom row, left panel) [108]. This activates EBP50/NHERF1/2 by releasing two PDZ domains from an autoinhibited state [109]. The two PDZ domains bind integral membrane receptors and channels, forming a scaffolding complex with the bound FERM protein at the membrane.

Merlin functions as an upstream activator of the Hippo signaling pathway [12]. It can do this either directly or indirectly by recruiting the kinase, Lats 1/2, to the membrane. The two Lats kinases each contain a short, FERM binding domain, N-terminal of the kinase domain. The crystal structure of merlin in complex with a Lats 1 peptide corresponding to the FERM binding domain shows that the Lats peptide binds as an α helix to subdomain F2 (Figure 7, bottom middle panel) [62]. This binding mode is identical to that of helix α1C of the merlin-ERM CTD.

Merlin also functions in the cell nucleus. It can suppress tumorigenesis by inhibiting the Cullin 4-RING E3 ubiquitin ligase CRL4^DCAF1^ in the nucleus. This is mediated by the direct binding of merlin to DDB1-and-Cullin-4-associated Factor 1 (DCAF1). DCAF1 possesses an acidic C-terminal tail which binds to the merlin FERM domain. Crystal structures show that two regions from the DCAF1 tail bind to FERM either as a single β strand to subdomain F3, extending the β sheet at β5F3, as per the majority of membrane protein cytoplasmic domains [110] or by forming a β hairpin at the same site (Figure 7, lower panel, right) [111].

On surveying all the peptide:FERM complexes for merlin-ERM proteins (Figure 7), it is striking that the majority of these peptides bind to the FERM domain in an identical manner to the merlin-ERM CTD in the closed FERM:CTD complex (see structure of *Sf*moesin on the left of Figure 7 as a reference). Thus, all of these interactions would be blocked in the closed state of the appropriate merlin-ERM protein. Even the extension of the F1 β sheet by MT1-MMP would be blocked by the central coiled-coil domain (Figure 7: compare radixin:MT1-MMP (top right panel) to *Sf*moesin).

Other interaction partners include Dbl (a Rho-specific guanine nucleotide exchange factor). Large morphological changes to human breast carcinoma cells (MDA-MB-231) have been linked to Dbl’s ability to be recruited to the plasma membrane when mutant forms of ezrin are expressed (E244K) [112]. This mutation disrupts a major salt bridge between F3 and the CTD and possibly binding partners. For example, the crystal structure of radixin with EBP50 reveals the very same salt bridge utilization to stabilize the interaction [108].

## 6. Post-Translational Modifications (PTM) of Merlin-ERM Proteins

The merlin-ERMs are highly dynamic proteins that undergo a myriad of post-translational modifications. (A good collation of PTM data from multiple sources can be found under each protein’s name at the post-translational modification database PhosphoSitePlus [113]). As expected with proteins that show strong conservation there are analogous modifications at some conserved sites in all members of the family, although there are also many paralog specific modifications that likely give rise to specific variations in function. This section compares and contrasts the known post-translational modifications in a structural setting. Numbering of the proteins is based on the following UniprotKB references [114]—Merlin:P35240, Ezrin:P15311, Moesin:P26038 and Radixin:P35241 (see Figure 3). Phosphorylation sites discussed in this review are marked with black stars in Figure 3.

### 6.1. Phosphorylation

By far, phosphorylation is the most studied post-translational modification observed in the merlin-ERM family of proteins. How merlin-ERM proteins switch between open and closed conformations is not entirely resolved, but it does appear that phosphorylation at least contributes to the exchange between the two forms. All merlin-ERM proteins possess multiple phosphorylation sites, the most well-documented are discussed below. The merlin-ERM proteins are well known to be the targets of numerous kinases and are phosphorylated at numerous sites [103,115,116,117,118,119,120]. There are at least 36 separate sites reported for moesin, 34 for ezrin, 15 for radixin and 11 for merlin. Of these numerous reported phosphorylation sites only two are observed at conserved sites in both merlin and at least two of the ERM proteins suggesting that phosphorylation sites are poorly conserved.

#### 6.1.1. Common Phosphorylated Merlin-ERM Residues

Only one conserved phosphorylation site in the merlin-ERM proteins has been identified via high throughput proteomic studies: Thr477 in merlin and Thr/Ser468 in the ERM proteins, including Sfmoesin (see Figure 3). This residue is located at the C-terminus coiled-coil helix α3H, immediately prior to the polyproline region. It seems unlikely that the phosphorylation of this residue (merlin Thr477, ezrin Thr468, moesin/radixin Ser468) is affected by the polyproline region as moesin does not have a polyproline region. We note that there is an additional phosphorylation site in the coiled-coil residing in the hinge region: Thr419 in merlin and Ser413 in ezrin (Thr413 in moesin but Asn414 in radixin). Cellular and biochemical data regarding all of these phosphorylation sites is largely absent.

#### 6.1.2. Merlin Specific Phosphorylation—Serines and Threonines

Many phosphorylation sites in merlin have been identified by multiple independent studies, using both low and high throughput methodologies and a large number of these sites are conserved in merlin across species suggesting they are biologically relevant. In particular, the following residues have been identified by 10 or more separate studies: Ser10, Ser13, Ser518 and Thr581.

Ser10 and Ser13 arise in the specific merlin N-terminal extension that precedes the FERM domain (see Section 8.1) [121]. Ser10 is phosphorylated by protein kinase A (PKA) in vivo, sharing a common kinase partner with Ser518 (see below). To assess the role of phosphorylation at Ser10, an S10D phosphomimetic mutant was constructed and shown to affect normal morphology of the actin cytoskeleton [122]. More specifically, for merlin knockout *NF2*^−/−^ mouse embryonic fibroblasts, the phenotypic difference between WT and S10D transfection was the formation of dense filopodia-like structures in S10D. The unphosphorylatable S10A mutant also showed defects in cell migration and lamellipodia formation. It has been shown that Ser10 is phosphorylated by all Akt (protein kinase B) isoforms in vitro and in vivo, resulting in recognition by the proteasome for degradation [123]. There is no specific information regarding the role of Ser13 (it was identified by a high-throughput mass spectrometry study) but considering its proximity to Ser10 it may play a similar role.

The most reported post-translational modification for merlin is phosphorylation at Ser518. It has been observed in human, mouse and rat proteins, however this residue is not conserved in the ERM proteins (it is equivalent to ezrin Asp510 and either moesin Lys501 or Arg503 and radixin Asn507 or Arg509, depending on sequence alignment, which is poor in this region). Merlin Ser518 is in the CTD. The crystal structure of the merlin:CTD complex containing the phosphomimetic mutation S518D (along with the structural A585W mutation), residues Thr512 to Ile546 form an extended α helix, N-terminally elongating the first of the four α helices at the C-terminus of the CTD (α1C). This extended α helix may result from crystal packing (see Section 4.3).

The formation of an extended α helix in this region differs from the ERM full-length model based on the crystal structure of *Sf*moesin where the residue equivalent to Ser518 (*Sf*moesin Thr461) resides after the β-strand that extends the FERM subdomain F3 β sheet but before the C-terminal α helical region (Figure 8d shows a model of merlin Ser518 based on a full-length *Sf*moesin structure). In the ERM case, this residue (equivalent to merlin Ser518) lies opposite the highly conserved ^308^MRRRK^312^ motif at the C-terminus of α1F3 in subdomain F3 (Figure 8d, shown in red). Based on the ERM model phosphorylation of Ser518 would likely destabilize the FERM–CTD interaction; however, studies assessing this are conflicted.

Studies of Ser518 mutants reveal complicated biological phenotypes from which it is difficult to infer biological function. For example, when the merlin phosphomimetic S518D mutant protein is expressed in human bladder-derived epithelial cells (RT4 cells) dramatic filopodial extensions form [115]. These filopodial extensions are not observed when either WT or S518A mutant merlin is expressed in the same cells, suggesting that Ser518 phosphorylation may be important for the formation of filopodia [115]. Phosphorylation at Ser518 appears to change the affinity of the FERM–CTD interaction [124,125], although data are conflicting. Phosphorylation of Ser518 does not affect merlin’s thermal stability [62,126] and isothermal calorimetry experiments indicate that the S518D mutation does not affect the binding constant for the FERM–CTD interaction (both WT and S518D having K_D_ ~4 µM) [62]. Small-angle neutron scattering (SANS) data indicate that in the presence of PI(4,5)P_2_, WT merlin adopts an open conformation whilst the S518D mutant is closed [127]. Recently it has been proposed that lipid binding is the sole contributor to the regulation of FERM–CTD interaction in merlin [70].

If Ser518 is not affecting the FERM:CTD binding affinity and thus regulating the open-closed transition, what else might it be doing? It has been proposed that Ser518 phosphorylation is an important switch in the regulation of the Hippo pathway. Merlin is a known upstream activator of the Hippo pathway kinases Lats 1/2. It functions by binding Lats 1/2 (see Figure 7) and recruiting these kinases to the membrane where they can activate subsequent complexes. Data from structural studies has revealed that the Lats 1/2 binding site on merlin’s FERM is blocked by merlin’s CTD (see Figure 7). When angiomotin (a protein with roles in cell motility and angiogenesis) binds merlin, the CTD is released and the Lats 1/2 binding site becomes available. Phosphorylation of Ser518 prevents angiomotin from binding merlin, which in turn ensures the CTD is masking the Lats 1/2 binding site on merlin’s FERM and thus inhibits Hippo kinase activation [62]. The kinase PAK2 (a Rac effector kinase) has been shown to phosphorylate Ser518 [124].

Merlin isoform 1 is also phosphorylated at Thr581, which is located at the C-terminus of the penultimate α helix, α3C, in the CTD (Figure 3 and Figure 8e, red spheres). This C-terminal region is missing in all other isoforms of merlin, however, isoform 1 bears the closest relationship to the ERM family and all invertebrate merlins, each having an identical C-terminal α helical structure (see Section 3.2 and Section 4.3). Additionally, merlin isoform 1 is the most abundantly expressed isoform. The phosphorylation of Thr581 has been reported in numerous high-throughput mass spectrometry analyses of various cancer cell lines; however, it is not well characterized. Thr581 lies adjacent to the highly conserved merlin-ERM salt bridge between the FERM and CTD domains (merlin Glu260–Arg588, Figure 8e), hence, phosphorylation of Thr518 is likely to alter the electrostatics, potentially weakening the FERM–CTD interaction. We note that merlin Thr581 corresponds to ezrin Arg572, which is conserved in most ERM proteins (with some Arg to Lys substitutions in invertebrate ERM sequences).

#### 6.1.3. ERM Phosphorylation

There is much greater conservation between the phosphorylation sites within the closely related ERM paralogs with at least 5 phosphorlyation sites being strictly conserved and phosphorylated in all three paralogs (see PhosphoSitePlus [113]). These resdiues are 1: ezrin-Ser66/radixin-Thr66/moesin-Thr66, 2: ezrin-Tyr270/radixin-Tyr270/moesin-Tyr270, 3: ezrin-Tyr291/radixin-Tyr291/moesin-Tyr291, 4: ezrin-Ser535/radixin-Ser532/ moesin-Thr526, and 5: ezrin-Ser536/radixin-Ser533/moesin-Ser527. Much of this data has been reported by large high-throughput mass spectroscopic analyses and there is currently little data of the specific role of each phosphorylation site. However, in all these cases phosphorylation has been observed multiple times and for more than one organism suggesting these sites are relevant to function (see PhosphoSitePlus [113]).

The first two sites (ezrin Ser66 and Tyr270) are near the putative FERM:membrane interface. Ser66 is in a single turn helix in subdomain F1 (Figure 8a) while Tyr270 is on β strand β7F3 of the outer β sheet in subdomain F3. The phosphorylation of these two residues could affect the FERM–membrane interaction. The remaining sites (ezrin Tyr291, Ser535 and Ser536) are at the FERM:CTD interface and phosphorylation may weaken the FERM–CTD interaction favoring the open form. Tyr291 is near the C-terminus of the α helix in subdomain F3 and its hydroxyl moiety interacts with the region of the CTD linking the β strand to the terminal α helical region. Ser535 and Ser536 lie on α helix α1C in the CTD. The former is in close proximity to Glu120 in FERM subdomain F2, thus, phosphorylation of these two serines may weaken the FERM–CTD interaction.

There are other notable residues for each specifc ERM paralog that are repeatly identified as phosphorylated (see PhosphoSitePlus [113]). These include for ezrin: tyrosines 116, 146, 354, 424, 478, 483 and 499, threonines 332, 425 and 567, and serines 366, 482 and 539. Only one unique site in radixin has been highly reported (tyrosine 55). Similar to ezrin, moesin has numerous highly phosphorylated residues including: serines 74, 384, 407, 429, 527, 576, tyrosines 116 and 556 and threonine 425. Ezrin Tyr146 is noteworthy as it is conserved in vertebrate ERMs and it resides in a long loop in subdomain F2 connecting helices α3F2 to α4F2 (Figure 8c). The tyrosine side chain points in towards the body of the protein and in the crystal structure of the ezrin FERM, the side chain maintains order, remaining visible in the electron density, while the backbone is not ordered in this loop [56].

All merlin-ERM proteins contain a pair of phosphorylatable (usually threonine) residues that are spatially juxtaposed in van der Waals contact on the FERM:CTD boundary (Figure 8b; see PhosphoSitePlus [113]). In the loop between β strands β3F3 and β4F3 of subdomain F3 of the FERM domain, lies a threonine residue (merlin T251, ezrin T235, radixin T235, moesin T235) that is conserved in all merlin-ERM proteins with the exception of nematode ERMs, where it is conservatively replaced by serine. In van der Waals contact of this residue (within ~4 Å) lies a completely conserved threonine (merlin T576, ezrin T567, radixin T564, and moesin T558) on the penultimate α helix (α3C) of the CTD. The phosphorylation of this second threonine has been reported across numerous species and all three ERM paralogs [103,118,128]) and there is much data regarding biological phenotypes for phosphomimetics at this site. Phosphorylation of this residue is associated with activation of ERM proteins, i.e., the separation of the FERM and CTD domains. However, phosphorylation has not been reported for the corresponding merlin residue Thr576 [115]. Little has been reported regarding the phosphorylation of the first threonine (merlin T251, ezrin T235, radixin T235, moesin T235); however, for ezrin this residue has been reported to be phosphorylated in mouse by CDK5 [129].

This pair of residues is so highly conserved and associated with phosphorylation that they are conspicuous. If both residues were phosphorylated then due to the close proximity of the phosphates they would likely repel each other causing significant structural alterations. Given that the phosphorylation of the CTD residue (ezrin T567, radixin T564, and moesin T558) is already associated with destabilizing the FERM:CTD complex for ERM proteins, the concomitant phosphorylation of the FERM threonine (ezrin T235, radixin T235, moesin T235) would likely enhance the effect. Surprisingly, although this pair of threonines has been conserved in the merlin family, no phosphorylation has been observed to date [115]. There is now growing evidence that alternate phosphorylation mechanisms and/or secondary phosphorylation sites enable ezrin, radixin and moesin (along with merlin) to be individually regulated and/or work in concert (see for example [130,131]).

Key insights into the role of phosphorylation at the residue equivalent to ezrin Thr567 have been gained from the use of phosphomimetic constructs (T567D and T567E) and non-phosphorylatable mutants (T567A). Indeed, introduction of an ezrin T567D mutation induces the formation of striking membrane ruffles and tufts of microvilli on the cell surface [132]. Similarly, an ezrin T567E mutation has been shown to increase membrane tension, slow migration, and impede endothelial transmigration of lymphocytes in vivo in mice [133]. Experiments suggest that phosphorylation at this site may increase ERM protein affinity for PI(4,5)P_2_ [134] and affinity for binding actin; however, it is not yet clear whether the increase in affinity for actin is due to a direct increase in binding affinity at the actin-binding site, or if phosphorylation at this position reduces the FERM:CTD complex affinity making the actin binding domain more accessible [68,69,135]. Experiments assessing non-phosphorylated and phosphorylated C-terminal domains of radixin at Thr564 (equivalent to ezrin Thr567, moesin Thr558) showed no significant difference in their actin binding ability, implying that phosphorylation does not regulate actin binding [118]. Further experiments of CTDs showed that the phosphorylated Thr564 radixin was remarkably suppressed with regards to FERM interaction affinity.

### 6.2. Cysteine Modifications: S-Nitrosylation of the Conserved F3 Loop

A striking feature shared by the merlin-ERM proteins is the invariable conservation of two cysteine residues in the FERM domain (marked by red stars in Figure 3). The first conserved cysteine (merlin Cys133, ERM Cys 117) is located in subdomain F2 in a loop joining α-helices α2F2 to α3F2 (Figure 3). This loop is surface accessible in the FERM only structure (putative open form), albeit with the cysteine side chain pointing into the core, however, it is completely buried in the FERM:CTD complex. The second conserved cysteine (merlin Cys300, ERM Cys284) lies at the center of the F3 α-helix, α1F3 (Figure 3). Its side chain is buried within the core of subdomain F3 in the FERM only structure with accessibility further decreased in the FERM:CTD complex due to the overlaying of the additional β strand formed by the CTD against the outer β sheet in FERM subdomain F3. It is curious that these two cysteines have been strictly conserved in the merlin-ERM family given that merlin and the ERM proteins diverged about 0.5 billion years ago. Additionally, these two cysteines are conserved in the FERM domains of erythrocyte band 4.1, the related DAL-1/band 4.1B proteins and tyrosine-protein phosphatase non-receptor type 3 proteins. For the more divergent FERM domains such as those in the E3 ubiquitin-protein ligase MYLIP and MIR (a myosin light chain interacting protein), only the first cysteine is strictly conserved. Finally, for other FERM domains both cysteines are absent, such as Talin-1, the JAK (tyrosine-protein kinases), focal adhesion kinases (FAK)/proline-rich tyrosine kinases, the Pleckstrin homology domain-containing family H proteins and the unconventional myosin-VIIb protein.

S-nitrosylation is a ubiquitous post-translational modification of cysteines that is emerging as a principle mechanism for nitric oxide (NO)-mediated signal transduction critical for cellular function [136]. S-nitrosylated species of both ezrin and moesin have been identified by proteomic analysis [137]. Jia et al. have also demonstrated that a I/L-X-C-X_2_-D/E motif centered around Cys117 of moesin is necessary and sufficient for specific S-nitrosylation via a stimulus inducible NO synthase complex (iNOS-S100A8/A9). Note that Cys284 has not been observed to be S-nitrosylated. Along with the conserved merlin Cys133/ERM Cys117, this I/L-X-C-X_2_-D/E motif is conserved in all merlin-ERM proteins suggesting that S-nitrosylation may occur across the family [136,137,138]. It is possible that such post-translational modifications would alter the FERM–CTD interaction as merlin Cys133/ERM Cys117 lies directly beneath α helix α2C of the CTD.

### 6.3. Ubiquitination and Acetylation of Merlin-ERM Proteins

Like numerous other proteins, merlin-ERM proteins have multiple reported ubiquitination [139,140] and acetylation [141,142] sites including many sites that are reported to have both modifications. Ubiquitination occurs most commonly on lysine resdiues; however, cysteine, serine, threonine or the N-terminus of a protein have also been reported [143,144,145]. In the case of the merlin-ERM family, all modifications observed to date are on lysine residues. Likewise, acetylation occurs predominately on the N-terminus of a protein, but also can occur on lysine residues.

Only two ubiquitinated or acetylated sites are conserved across the whole merlin-ERM family, being equivalent to ezrin Lys143 and Lys253. A larger number of ubiquitinaton/acetylation sites are conserved within the ERMs—these include lysine 3, 35, 72, 79, 83, 139, 162, 209, 211, 237, 254, 306 and 458. It is notable that the majority of these sites arise in the FERM domain. Mapping these conserved lysine modification sites onto the crystal structure of ezrin shows distinct grouping of the sites on areas that have distinct, basic surface charges (Figure 9). Both ubiquitination and acetylation modifications would reduce surface charges. While the direct role of the charged surfaces on the ERM protein is unknown, it is likely that the charged surfaces are important for maintaining protein-protein interactions or protein-membrane interactions. In light of this it seems possible that ubiquitination and acetylation in these areas may play a role in modulating such interactions.

Ezrin had been reported to interact directly with the E3 ubiquitin ligase WWP1 via the ^475^PPVY^478^ consensus motif (see Figure 3), located at the end of the ezrin polyproline tract at the start of the CTD [139]. While it is not clear whether ezrin phosphorylation is absolutely necessary for WWP1 ubiquitination [139], intuitively, the ^475^PPVY^478^ binding motif must be accessible for binding the WWP1, hence, it is likely to be released from the FERM domain favoring an open form of ezrin, which would expose the ubiquitination sites, most of which are located in the FERM domain. Interestingly, WWP1 ubiquitination does not appear to target ezrin for degradation [139], thus ezrin may act as an adapter protein recruiting further substrates for WWP1 as part of a regulatory mechanism. Merlin has also been reported to have direct roles in controlling ubiquitination and is known to translocate to the nucleus and directly inhibit the CRL4(DCAF1) E3 ubiquitin ligase [140].

### 6.4. Sumoylation of Merlin-ERMs

Sumoylation is a relatively rare modification within the merlin-ERM family. Merlin has been observed to be sumoylated on lysine 15, 76 and 550. Sumoylation at Lys76 is required for merlin’s tumor-suppressive acitivity in tissue culture models although the mechanism for this remains to be elucidated [146]. There is little data for the family of ERM proteins, however, the single ERM protein from *Drosophila*, called Dmoesin, has also been observed to be sumoylated (precise site unidentified), again the role and mechanism is unknown [147].

### 6.5. Succinylation

The 100 dalton succinyl group is a large modification added to lysine residues that significantly changes the lysine side chain charge from +1 to −1, thus on surfaces involved in ionic interactions, the addition of a succinyl group would be of great significance. Succinylation is reported to occur on ezrin Lys60 [148], and a global proteome analysis for human gastric cancers also identified Lys438 in ezrin [149], human radixin Lys435, mouse radixin Lys83 [142], and moesin Lys79 and Lys165 [148]. The biological relevance of these modifications is unknown.

### 6.6. ADP-Ribosylation of ERM

ADP-ribosylation is the conjugation of ADP-ribose groups (one or more) to a protein. Attachment of the ADP-ribose group occurs generally on arginine side chains. ERM proteins are ADP-ribosylated by the *Pseudomonas aeruginosa* virulence factor and cytotoxin ExoS [150]. This is a possible pathogenic mechanism whereby post-translational modification of the ERM protein by the bacterial toxin improves infection. This modification was originally identified on moesin but further testing revealed it also occurs on radixin and ezrin [151]. The arginine residues of moesin that are ADP-ribosylated are Arg553, Arg560 and Arg563, which span the penultimate α helix in the CTD (α3C). These arginine residues are conserved in vertebrate ERMs and they cluster around Tyr565, which is a reported phosphorylation site. The ADP-ribosylation was shown to inhibit phosphorylation at this residue [150]. The ADP-ribosylation of these three arginine residues is likely to disrupt the structure of the FERM:CTD complex, suggesting that the closed monomer and dimer structures would not be able to form. Additionally, the ADP-ribosylated CTD may not be able to interact with actin.

From the example above and the knowledge that a number of viruses require ERMs for their infection process (see Section 3.2), it seems that ezrin is important for many pathogens entry into cells. There are several reported cases of knock-downs of ERM proteins that block infection (for example [39,152]). Furthermore the phosphorylation state of ezrin has been shown to be important in this process. For example the infection of HIV-1 via CXCR4 co-receptor requires phosphorylated ezrin [153]. Finally, some viruses interact specifically with the protein. For example the SARS coronavirus spike protein interacts with ezrin’s FERM domain via the F1 lobe [152]. Thus bacteria and viruses interact specifically with ezrin (or upstream in the ezrin activation pathway (for example [154])) to target and modulate ezrin activation. ADP-ribosylation by *P. aeruginosa* is one instance where the pathogen directly modulates ezrin’s post-translational modifications. It seems likely that other pathogens including viruses may specifically alter ezrin’s post-translational modifications.

## 7. Common Structural and Functional Characteristics of Merlin-ERM Proteins

### 7.1. Lipid Binding

Co-sedimentation assays, fluorescence correlation spectroscopy and lipid arrays have all shown merlin-ERM proteins specifically bind phosphoinositides in vitro, with both ezrin and merlin reported to have the highest affinity for PI(4,5)P_2_ [155,156,157,158,159].

Several lines of evidence suggest merlin-ERM proteins also interact with PI(4,5)P_2_ in vivo. The overexpression of PtdIns4P 5-kinase α (PIP5Kiα), an enzyme which produces PI(4,5)P_2_ (which should increase the concentration of PI(4,5)P_2_ in the membrane) results in the increased recruitment of ERM proteins to vesicular membranes [160]. Enzymatic hydrolysis of PI(4,5)P_2_ by the activation of phospholipase C reduces ERM localization at the membrane [103,160,161]. Finally, ERM proteins are re-localized from the membrane to the cytoplasm when cells are treated with neomycin (a drug which binds PI(4,5)P_2_ effectively reducing the concentration in the membrane [160]).

Merlin-ERM proteins bind PI(4,5)P_2_ through their FERM domain [58,155,159,162], although the precise structural location of the PI(4,5)P_2_ binding site/s are poorly resolved and may differ between the merlin-ERM paralogs. The PI(4,5)P_2_ affinity of the FERM domains of ezrin and moesin decrease with increasing ionic strength, indicating the interaction is largely ionic in nature, although ezrin binds more strongly to PI(4,5)P_2_ containing-vesicles than to other vesicles with similar charge, suggesting that the interaction with PI(4,5)P_2_ is specific [157,163]. Fluorescence anisotropy assays that can detect the insertion of proteins into the membrane, show that neither ezrin or moesin insert deeply into the membrane [163]. Together these data suggest merlin-ERM proteins bind to just the polar head group of PI(4,5)P_2_ through specific electrostatic interactions and do not deeply insert into the lipid bilayer.

Two potential lipid binding sites, termed the ‘pocket’ and the ‘patch’, have been identified across the merlin-ERM family based on mutagenesis and structural data (Figure 10a–c) [58,156,162,164]. Both sites occur on the same face of the FERM domain, spanning the F1 and F3 subdomains. Lysine-to-asparagine mutations in these binding sites decrease merlin-ERMs affinity for PI(4,5)P_2_ and localization to the membrane [103,156,162]. A mutation in merlin within the pocket (K79E) is associated with neurofibromatosis type 2, a dominant disease that can lead to the development of neuronal tumors [60], demonstrating that PI(4,5)P_2_ binding is essential for merlin function. Molecular dynamics simulations of moesin FERM indicate both sites could bind the membrane simultaneously [163].

Interestingly, a recent crystal structure of the merlin FERM domain bound to a PI(4,5)P_2_ short chain derivative showed the lipid can bind a nearby region between the F2 and F3 subdomains. This region was termed the ‘pouch’ and is different to the previously identified pocket and patch sites (Figure 10f). Mutation of this site resulted in a loss of lipid-binding and merlin-dependent cell growth suppression [70]. Five of the eight residues involved in binding PI(4,5)P_2_ in this region are conserved in all the ERM proteins, with two conservative substitutions. This data suggests merlin-ERM proteins may interact with PI(4,5)P_2_ using multiple binding sites.

While PI(4,5)P_2_-binding of the merlin-ERMs does not cause large changes to lipid structure, it can bring about conformational changes in the merlin-ERM proteins [55,162]. The full-length *Sf*moesin crystal structure shows that two sections of the CTD obscure the putative membrane-binding sites: the patch, the pouch and the pocket (compare full-length *Sf*moesin, Figure 10a, with FERM only, Figure 10b). In particular, the region just after the central helical domain passes over the pouch and pocket sites (Figure 10a, red linear region on right leading from the helical domain, orange), additionally, a β strand, β1C runs across the face of the membrane-binding surface (Figure 10a, red linear region bisecting the FERM). These segments of the CTD also disrupt the large, positively charged surface of the FERM that presumably binds the negatively charged lipid surface (compare Figure 10d to Figure 10e). Finally, docking the full-length *Sf*moesin structure to a cartoon representing the lipid bilayer with the putative lipid binding face parallel to the membrane shows that the coiled-coil of the helical domain points into the membrane, which is unphysical (Figure 10h). Thus, membrane binding must involve a significant structural rearrangement compared to the reported full-length *Sf*moesin structure.

Indeed, several lines of evidence suggest binding of PI(4,5)P_2_ causes a more open merlin-ERM monomer structure (note: PI(4,5)P_2_ refers to natural, long chain PI(4,5)P_2_, unless stated otherwise). Ezrin and moesin display tryptophan quenching and an increased susceptibility to proteolysis in the presence of PI(4,5)P_2_ [155,165,166]. Isothermal titration calorimetry experiments of PI(4,5)P_2_ in solution have shown that the binding of full-length ezrin is more entropically favorable than that of the ezrin FERM domain [165]. SANS experiments have suggested both ezrin and merlin form elongated monomers in the presence of PI(4,5)P_2_ [127,167]. Finally, the recent structure of merlin FERM bound to the soluble, short chain PI(4,5P)_2_diC_8_ contains a region of the central α-helical domain that undergoes a 60° rotation in the presence of the lipid, and exchange experiments (using deuterated solvent) reveal that the interacting faces of the merlin CTD and FERM are more solvent exposed in the presence of the soluble, short chain PI(4,5)P_2_diC_8_ [70]. Cumulatively, these data suggest PI(4,5)P_2_ binding results in the formation of open merlin-ERM monomers by inducing a conformational change that causes the release of the CTD from the FERM domain. This is the current most favored model regarding ERM protein activation. Consistent with this model is the data showing an increase in ezrin and moesin homodimers in the presence of PI(4,5)P_2_ by gel filtration and analytical ultracentrifugation [95,167]. Merlin-ERM membrane localization is not just dependent on lipid binding, but also involves interactions with other proteins. The presence of PI(4,5)P_2_ increases the affinity of both ezrin and merlin for the cytoplasmic tails of their integral membrane protein partners [99,159,168,169]. This is consistent with the fact that the complexes between merlin-ERM and their binding partners mimic the FERM:CTD complex (see Section 5 and Figure 7).

ERM proteins stabilize numerous different membrane structures, some of which have positive curvature (endosomes, intracellular vesicles) or negative curvature (filopodia and microvilli), while others (plasma membrane) are flat on the molecular scale. The extent to which ERM proteins shape membranes, in particular determine curvature, is unclear. A recent cryo-electron microscopy study has shown that both WT ezrin and the phosphomimetic mutant, T567, can self-associate so as to tether PI(4,5)P_2_ containing membrane vesicles to each other forming stacks of lamellar vesicles [96]. In these flat membrane structures, ezrin appears to be in the domain-swapped dimer form with the FERM domains interacting with the juxtaposed membranes (see Section 4.5). Ezrin dimers lie perpendicular to the membrane plane, forming a dense, brush-like structure through lateral self-association on the membrane. Both WT ezrin and T567D were able to bind to highly curved (positive curvature) galactocerebroside membrane nanotubes (~25 nm external diameter) with WT ezrin forming ordered stacks of nanotubes, while T567D produced disordered assemblies of nanotubes. These authors also showed that neither WT or T567D ezrin were enriched on PI(4,5)P_2_ containing membranes with negative curvature [96]. However, on addition of an I-BAR domain containing protein, IRSp53, which is known to both induce negative membrane curvature and bind ezrin, both WT and T567D ezrin were recruited to the region of curvature.

### 7.2. Interactions with Microtubules

The merlin-ERM proteins all interact with various components of the cytoskeleton, which allow merlin-ERM proteins to control cellular shape and structure. In this Section specific interactions with tubulin are discussed while interactions with actin are discussed in Section 8.2 and Section 8.3.

A direct interaction between merlin-ERM proteins and microtubules, mediated by the FERM domain has been shown using cosedimentation and microscopy based assays with the *Drosophila* ERM protein (*Dmoesin*), and human moesin and ezrin [95,170]. In cosedimentation assays the *Dmoesin* T559D phosphomimetic mutant (equivalent to ezrin T567D) showed significant reduction in binding microtubules whereas the constitutively open truncation mutant (encoding residues 1–559) experiences a >30-fold increase in binding. This result suggests that there is a steric masking of the binding site even post-phosphorylation of *Dmoesin* Thr559 (equivalent to ezrin Thr567 phosphorylation) [170]. Mutagenesis experiments have identified two lysines on the loop linking β strands β1F3 and β2F3 in the inner β sheet of FERM subdomain F3 (K212/213 in *Dmoesin*, K211/212 *Hs*ERM, K227/228 *Hs*merlin) as important for microtubule binding [170]. This loop forms the binding pocket for the extreme C-terminus of the CTD in the FERM:CTD structure. Consistent with these observations, the latter half of the merlin FERM domain (178–367) directly interacts with polymerized microtubules [171].

While there is evidence of a direct interaction between the merlin-ERM proteins and polymerized microtubules, indirect associations may also contribute to their co-localization. In vivo studies in *Drosophila* have revealed that merlin tracks bidirectionally along microtubules, and pull-down experiments revealed both kinesin-1 and cytoplasmic dynein as candidate transporters [172]. The *Drosophila* merlin T616D mutant (equivalent to Thr576 in human merlin) abolishes the transport of merlin, while further work indicates that phosphorylation of merlin regulates the binding of the motor proteins. This data is hard to reconcile with reports that the T576D post-translational modification had no effect on merlin’s function in humans and makes this area of research particularly intriguing.

## 8. Specific Structural and Functional Characteristic—the Differences between Merlin and the ERMs

There are a few specific characteristics unique to merlin and the ERM proteins that separate them from each other. This Section discusses these differences focusing specifically on the sequences and structure of the proteins that give rise to these qualities.

### 8.1. Merlin and the Nucleus

Merlin has been reported to shuttle between the plasma membrane, the cytoplasm and the nucleus, while the ERM proteins do not enter the nucleus. To achieve this merlin must be equipped with both nuclear import and export sequences, and also it seems likely there must be ways that these are regulated.

Indeed merlin has an experimentally determined non-canonical nuclear localization signal (NLS) sequence within the F1 subdomain of the FERM (^24^VRIV^27^) [173], and three functional nuclear export signals (NES)—two contained in the FERM domain (exportin 1 recognition sequences [174]) and one in the CTD (with a sequence of ^538^LNELKTEIEALKL^551^ [175]). These NLS/NESs are unique to merlin and do not appear to be present in ERM proteins. There is also a cytoplasmic retention factor encoded on exon 2 which encodes the second half of the F1 lobe of the FERM domain [103].

The functionality of these sequences have been demonstrated in vivo by deletion and site directed poly-alanine mutagenesis studies [175,176] which revealed that nuclear export of merlin is exportin 1 dependent and may be abolished by deletion of the exons containing consensus sequence. The deletion experiments must be interpreted with care as deletion of a part of the protein may cause significant disruption to the overall protein structure causing indirect effects. The poly-alanine substitution experiments also require care, as they may alter the structural integrity of merlin.

Finally, unique to merlin is an 18-amino acid N-terminal appendage (Figure 3). The structure of this region has not yet been resolved experimentally and it is not clear what, if any, secondary structure it possesses. Curiously, this region is predicted to be a consensus nuclear export sequence (NES) for exportin 1 recognition, and is significantly positively charged, however, its biological functional is not known.

### 8.2. C-Terminal ERM Actin-Binding Domain

One significant difference between the mammalian ERM proteins and merlin is a short (approximately 34 amino acid) motif within the CTD that binds F-actin that is present in the ERMs but absent in merlin [177,178]. The actin-binding motif was identified using truncated fusion proteins, and only binds filamentous F-actin, not monomeric G-actin [177]. It shows sequence similarity to the β subunit of CapZ, (an actin-capping protein) and to a proposed actin-binding site in myosin heavy chain [177]. Currently there is no high-resolution structure of the ezrin/actin interaction however work characterizing the interaction has revealed that residues Thr576, Lys577, Arg579 and Ile580 are intimately involved, with these residues forming a patch on the surface of the last helix of the ezrin structure [179]. These residues are strongly conserved between the ERM proteins.

Although this actin-binding motif appears to distinguish the ERMs from merlin for mammalian sequences, the picture is more complex when one considers the whole merlin-ERM family. For invertebrate sequences, the distinction between merlin and ERM actin-binding sequences is not clear and, looking at the sequences, one would conclude that both are likely to bind actin. More work is clearly needed to resolve this issue. A high-resolution structure of a complex between F-actin and an ERM CTD would shed much light on this interaction.

It is important to note that full-length ezrin does not bind F-actin in co-sedimentation assays [180,181], as the actin-binding site is occluded when the FERM and CTD domains interact (see Section 4.4 for the N- and C-terminal interaction). To activate actin binding, the FERM and CTD need to dissociate. Consistent with this, full-length ezrin when partially denatured with SDS is able to interact with actin [182]. Mutagenesis studies of Arg579 of ezrin (a residue involved directly in the actin binding site) have revealed this residue is also important in the interaction of the FERM and CTD. An alanine mutation at this site significantly decreases the interaction affinity of the FERM:CTD [179]. This conserved arginine (ezrin Arg579, merlin Arg588) forms a tight salt bridge with a conserved glutamic acid (ezrin Glu244, merlin Glu260) as discussed in Section 4.4.

Consistent with the idea that PI(4,5)P_2_ binding and phosphorylation contribute to the release of the shrouded actin-binding site, moesin isolated from F-actin in human platelets is exclusively phosphorylated at Thr558 [183]. Furthermore cooperativity of ezrin binding to actin is observed when both PI(4,5)P_2_ binds and Thr567 is phosphorylated [134]. Fluorescence microscopy revealed only PI(4,5)P_2_-bound ezrin interacts with F-actin, indicating the protein is likely in an open conformation upon binding to PI(4,5)P_2_ [184].

Recent confocal and cryo-electron microscopy studies have shown that both WT and the phosphomimetic T567D ezrin will form a brush-like coat on PI(4,5)P_2_ containing vesicles [96]. However, only vesicles coated with T567D bind to F-actin, which is consistent with the charged phosphomimetic mutation destabilizing the FERM–CTD interaction, allowing the CTD to bind F-actin. Cryo-electron microscopy images show that F-actin filaments lie parallel to the lipid bilayer, forming a layer that is separated from the outer leaflet of the membrane by the T567D ezrin assembly. The separation between F-actin and the outer leaflet of the membrane is about 25 nm [96], which is consistent with the length of the domain-swapped ezrin dimer [56]. Tsai et al., model the T567D ezrin layer as a lateral assembly of open conformation, extended ezrin monomers with the FERM domain bound to the membrane and the CTD bound to F-actin [96].

### 8.3. Merlin–Actin Binding

While the ERM proteins have the specific actin-binding domain in their C-termini described in Section 8.2 there is evidence that all merlin-ERM proteins can interact with actin via a second actin-binding site common to a number of FERM domain containing proteins. This interaction site includes residues 178–367 of merlin [171] and has been identified using in vitro direct protein-protein co-sedimentation assays with F-actin [185,186]. Similar experiments have shown that FERM domains from chicken talin and *Drosophila* myosin 7a can bind actin with low affinity [187,188]. The evolutionary conservation of the FERM–actin interaction supports a role of the merlin FERM domain in locating the protein to the actin cytoskeleton. Merlin is also able to bind to actin indirectly by interacting with both βII-spectrin and β-fodrin [189,190].

## 9. Discussion

There is still a lot we do not know about the structure of merlin-ERM proteins and how structure dictates function. Although we have a detailed structural picture of both isolated FERM domains and FERM:CTD complexes, to date, the only full-length high-resolution structure is that of the insect ERM *Sf*moesin in the closed monomeric state. More importantly, we do not have a true picture of the structure of the open form of merlin-ERM proteins. This may be difficult to obtain, as such a structure may only exist transiently in isolation, requiring binding partners to stabilize regions such as the central helical domain and the CTD.

The structure of the membrane-bound form of merlin-ERM proteins is also not well characterized. It is possible that with the advent of cryo-electron microscopy techniques, this state may be analyzed at near atomic resolution in the foreseeable future. Such a structure should resolve several key questions regarding structural transitions facilitating membrane binding. It may also shed light on the organization of merlin-ERM proteins on lipid bilayers. An important advance in this regard is the recent cryo-electron microscopy study of WT and the phosphomimetic, T567D forms of ezrin bound to PI(4,5)P_2_ containing lipid bilayers, which they tether, and T567D ezrin cross-linking the membrane to a layer of F-actin filaments [96] (see Section 4.5, Section 7.1 and Section 8.2). These images give a tantalizing picture of domain-swapped ezrin dimers extending from the bilayer in a perpendicular fashion while forming a dense, brush-like array of protein.

Comparing the structures of the ERMs to those of merlin shows a picture of tight structural conservation within the merlin-ERM family. At this point, structural differences between the merlin and ERM families appear to be subtle. FERM domain structures are superimposable (Figure 4c). FERM:CTD complexes are also near identical (Figure 4d), although the merlin FERM–CTD interaction may be weaker as crystallization required the A585W mutation to stabilize the FERM:CTD complex [62]. The differences observed so far between merlin and ERM proteins could all potentially arise from crystallographic packing concerns, for example, the N-terminally extended α helix α1C in the merlin:CTD complex [62], the unfurling of subdomain F2 in the merlin FERM [191] or the differences in binding of FERM domains to PI(4,5)P_2_ mimetics [58,70]. Taken together, the structural data indicate that there are no major differences between the merlin and ERM family proteins.

On examining the structures of the complexes between the merlin-ERM proteins and their binding partners, a common picture also emerges. For both merlin and the ERM proteins, the FERM domain appears to be a hub that binds an array of proteins largely via short peptides that adopt their structure on interacting with the FERM domain. Curiously, the structures of the peptides bound to the FERM domain largely mimic some portion of the CTD as seen in the structure of the FERM:CTD complex (Figure 7). This suggests that the CTD interacts with the FERM domain as a cover, masking the many different interaction sites in a way that mimics the FERM:binding partner complexes. Even the membrane-binding surface appears to be masked by a combination of the CTD plus the central helical domain (Figure 10).

There are a vast number of putative and confirmed post-translational modifications on the merlin-ERMs (Figure 3, Figure 8 and Figure 9). Many of these alter charged interaction surfaces on the FERM or CTD, thus, they are likely to alter the stability of either the FERM:CTD complex or the electrostatic FERM:membrane interaction (Figure 9 and Figure 10) or the affinity of merlin-ERM interaction with binding partner proteins (for example angiomotin and merlin Section 6.1.2). Very little is really known about the structural and dynamic effects of these PTMs.

One may conclude that the net result of specific post-translational modifications plus the binding of partner proteins, particularly integral membrane proteins, is to break the FERM:CTD complex and hence expose critical surfaces on the FERM domain to ensure stable membrane binding. Although attractive, this picture is not so straightforward. Several of the complexes (for example: moesin:crumbs, radixin:CD44 and merlin:DCAF1, Figure 7) result in binding-partner peptides covering the very surfaces of the FERM domain that would have been exposed on removal of the CTD in order to facilitate membrane binding (Figure 10). Thus, the association of the FERM domain with cell membranes may be more complicated and require new research approaches.

One idea is that the CTD constrains the structure of the FERM domain so as to prevent stable membrane binding (as opposed to just covering its surfaces). There is, indeed, indirect evidence that the structure of the FERM domain (and the CTD) differs in the open form when compared to the closed form. There are also small but significant differences between the crystal structures of lone FERM domains compared to FERM:CTD complexes [56]. However, as the complexes of binding partner proteins to the FERM mimic the FERM:CTD complex, it is not clear that the structure of the naked FERM is the same as that of the membrane-bound FERM or indeed the membrane-bound open form full-length merlin-ERM.

The importance of controlling the FERM–CTD interaction is highlighted by the isoforms of the merlin-ERM proteins. While ezrin and moesin appear to be expressed as a single isoform in mammals, both merlin and radixin have a complex set of isoforms. A common feature of these isoforms is the alteration of the merlin-ERM C-terminus. In merlin, the majority of alternative isoforms replace the C-terminus by one that is five residues shorter and more hydrophilic than that of the major isoform 1, which has a normal merlin-ERM C-terminus. This change is likely to alter merlin’s ability to form a stable FERM:CTD complex. In contrast, the majority of radixin isoforms have an additional 21 amino acids added to the C-terminus of the major isoform 1, which has a normal merlin-ERM C-terminus. The structural consequences of the radixin extension are difficult to predict. Isoforms of merlin and radixin await structural characterization.

Although interactions with the merlin-ERM FERM domain are well characterized, we still know very little about the interactions of the CTD with other proteins, particularly F-actin. This has been rendered difficult by the filamentous nature of F-actin. Again, the advent of high-resolution cryo-electron microscopy may shed light on CTD–F-actin interactions.

Given the structural similarity of merlin and the ERM proteins, how is it that these two protein families differ in cellular function? Their functional differences may come down to their specific interaction partners. It is possible that merlin and the ERMs represent essentially identical molecular systems, switchable protein interaction hubs that are coupled to distinct networks of binding partners where the differences between the ERM and merlin networks are dictated by specific differences between molecular interactions. Thus, evolution may have duplicated the primordial merlin-ERM switchable interaction hub and then specialized both merlin and ERM separately.

## 10. Speculation

What have we learned from the structural view of the merlin-ERM family? Merlin and the ERM proteins separated near the origin of metazoa, approximately 0.5 billion years ago. They diverged functionally, with the ERM associated with cell polarity and the maintenance of membrane structures, while merlin plays a role in contact inhibition, maintaining cell–cell junctions and signaling to the nucleus controlling cell proliferation. In a sense, merlin and the ERM have set up a set of coordinate axes in the cell: ERM associated with polarity perpendicular to an epithelial cellular layer and merlin associated with the lateral in-plane axis.

Despite this differentiation, merlin and the ERMs have preserved their structural similarity over this approximately 0.5 billion year time span. Their FERM and CTD domains appear to recognize each other sufficiently to form heteromeric complexes (Section 4.5). The central helical domain, which protrudes from the monomer structure as an elongated coiled-coil, has maintained both its length and the periodicity of the heptad repeat over its entirety including the phase of the repeat through the central hinge region (Section 4.3). These are unusual features to preserve over such a long time-frame unless there are specific selection pressures for maintaining these relationships between the distinct merlin and ERM families.

There are at least two possible sources of selection pressure that could have maintained the structural similarity between merlin and the ERMs: (i) the merlin and ERM proteins interact with each other, forming heterodimeric and hetero-oligomeric complexes, necessitating the maintenance of structural similarity; or (ii) merlin and the ERMs have preserved specific binding to some as yet unidentified partner protein complex(es) that require the preservation of their structural features. Given the extent of the structural conservation covering a large portion of the merlin-ERM structure, the recent cryo-electron microscopy data showing membrane-bound ezrin assemblies consistent with dimer structures [96] and that heterodimers of merlin-ERM proteins have been isolated from in vivo sources in multiple reports (Section 4.5 [93,97,98]), we favor the former interpretation: that merlin and the ERM proteins are two sides of the same coin, functioning as a unit by directly interacting with each other to effect their different functional roles.

## Figures and Tables

**Figure 1 ijms-20-01996-f001:**
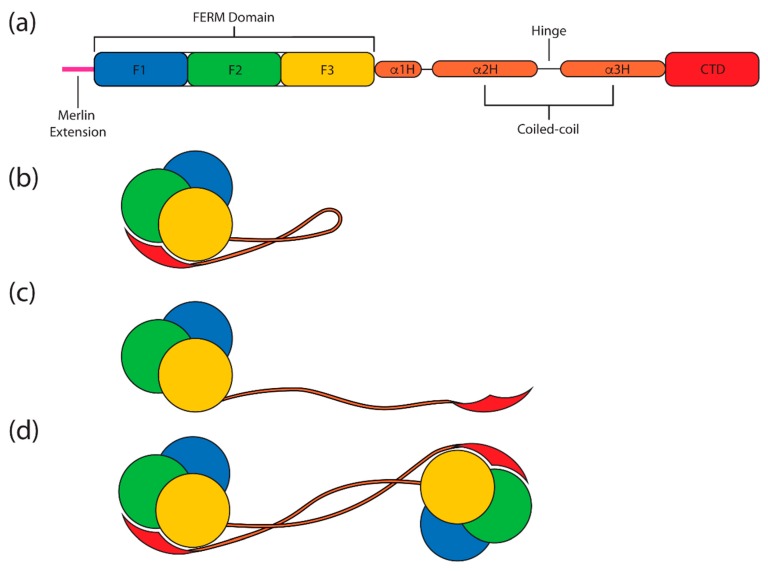
Schematic overview of merlin-ERM (ezrin, radixin and moesin) protein domain structure and states. (**a**) The conserved domain structure of merlin-ERM proteins. The N-terminal FERM (band 4.1 protein, ezrin, radixin, moesin) domain comprises three subdomains, F1 (blue), F2 (green) and F3 (yellow). This is followed by a central helical domain comprising three α helices (orange), with the latter two, α2H and α3H, forming a coiled-coil in the monomer structure. At the C-terminus lies the largely α helical C-terminal domain (CTD, red). Note that merlin contains an N-terminal extension that is not seen in the ERM proteins (magenta). (**b**–**d**) show various states of merlin-ERM proteins. (**b**) represents the closed state monomer structure where the CTD and FERM domains form a globular structure with the α helical coiled-coil protruding. (**c**) represents the putative open state, where an extended “helical” domain separates the FERM and CTD domains. (**d**) shows the domain-swapped dimer state.

**Figure 2 ijms-20-01996-f002:**
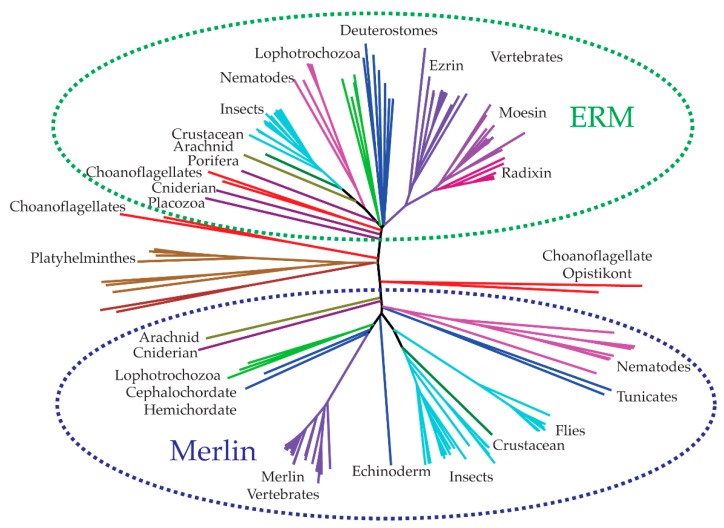
Phylogenetic tree demonstrating the divergence of merlin from the ERM proteins. The upper arbor shows ERM proteins spanning early branching metazoa through to vertebrates. The vertebrate ERMs diverge into distinct ezrin, radixin and moesin clades (top right). The lower arbor shows merlin proteins again spanning early branching metazoa through to vertebrates. Near the center of the tree lie various merlin-ERM proteins where it is not clear that they can be classified as either merlin or ERM from sequence analysis alone. We note that there are multiple choanoflagellate proteins (red branches) that appear to represent distinct merlin and ERM proteins. Sequences were obtained from the National Center for Biotechnology Information (NCBI, US National Library of Medicine) non-redundant protein database using the program BLAST [14]. Multiple sequence alignment and neighbor-joining phylogenetic tree were constructed with the program MUSCLE [15] using the EMBL-EBI webserver. The tree was drawn using the program FigTree v1.4.2 [19].

**Figure 3 ijms-20-01996-f003:**
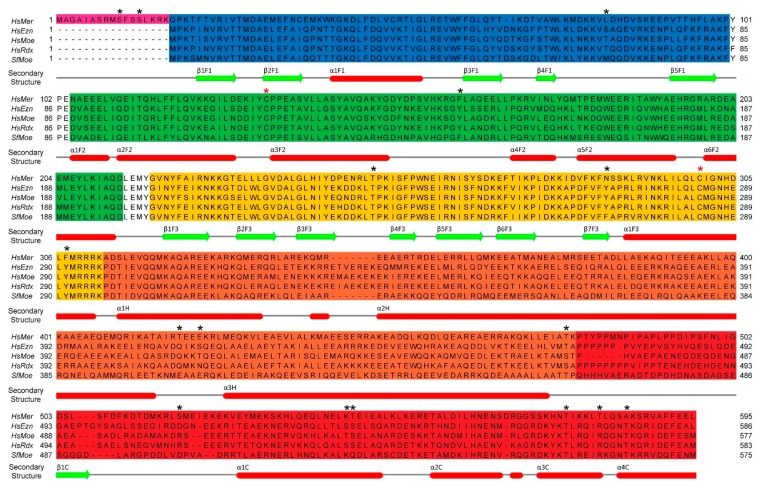
Multiple sequence alignment of human merlin-ERMs plus *Sf*moesin. Domains are colored as per Figure 1 (FERM subdomain F1—blue, FERM F2—green, FERM F3—yellow, Helical domain—orange, CTD—red). Secondary structural elements are shown based on the *Sf*moesin crystal structure. Black stars denote phosphorylation sites discussed in the text. Red stars indicate the two conserved cysteine residues. Sequence alignment was carried out with the program MUSCLE [15] using the EMBL-EBI webserver. The figure was prepared using the program Jalview [21].

**Figure 4 ijms-20-01996-f004:**
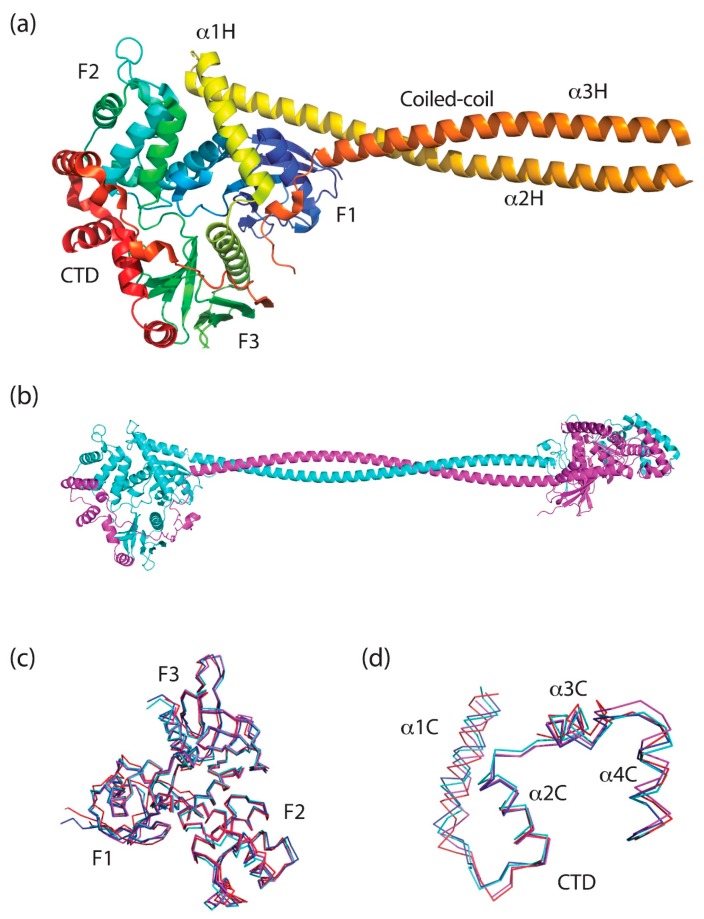
The structure of merlin-ERM. (**a**) The crystal structure of the only full-length ERM, *Sf*moesin, from *Spodoptera frugiperda* (PDB 2I1K) [55]. The protein is colored as a rainbow from the blue N-terminus to the red C-terminus. A striking feature is the coiled-coil extending from the globular FERM:CTD complex. (**b**) Model of full-length human ezrin domain-swapped homo-dimer which is based on the crystal structure of the ezrin FERM:CTD complex plus small-angle x-ray scattering data [56]. (**c**) Wire overlay of FERM domains (from crystal structures of FERM:CTD complexes of human merlin (4ZRJ, red) [62], ezrin (4RM9, Blue) [56] and moesin (1EF1, cyan) [61] and *S. frugiperda* ERM (2I1K, magenta) [55] displaying almost identical tertiary structure. (**d**) Wire overlay of the final α helical section of the CTD taken from crystal structures of FERM:CTD complexes (same structures as (**c**)), further displaying the tertiary similarity. This figure was made using the program PyMOL [63].

**Figure 5 ijms-20-01996-f005:**
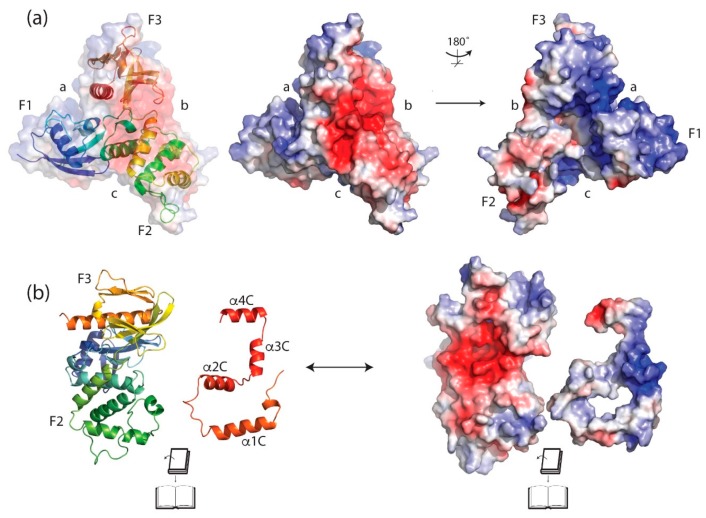
Surface electrostatics of the human ezrin. (**a**) Surface electrostatics of the ezrin FERM domain (PDB 4RMA, [56]). Left panel shows the protein backbone (ribbon diagram) overlayed with a transparent electrostatic surface. The clefts between FERM subdomains are labeled a, b and c. Middle panel shows the electrostatic surface in the same view, while the right panel is rotated 180° about a vertical axis in the plane of the page. The cleft between FERM subdomains F2 and F3 (labeled b) shows a large area of negative charge (red, middle panel) while the cleft between subdomains F1 and F3 (labeled a) shows a large surface of positive charge (blue, right panel). (**b**) The interface between the FERM and CTD complex is visualized by separating these domains and rotating the CTD by 180° about a vertical axis in the plane of the page (left panel). The right panel shows the electrostatic surface potential calculated for the separated FERM and CTD domains. It is clear that charge complementarity is important in stabilizing the FERM:CTD complex. Image rendered in PyMOL [63] showing Poisson–Boltzmann electrostatic potential calculated with PDB2PQR [66].

**Figure 6 ijms-20-01996-f006:**
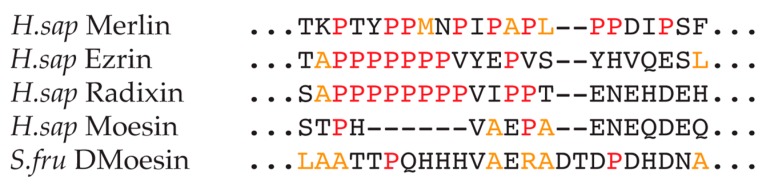
ClustalOmega sequence alignment of the proline-rich/polyproline region at the start of the CTD (just after the end of the coiled-coil in the helical domain). The alignment contains human merlin-ERM proteins and the insect ERM (*Sf*moesin) from *S. frugiperda.* Prolines highlighted in red and other residues more commonly found in Type II poly-proline helices highlighted in orange. The alignment was carried out using Clustal Omega [80].

**Figure 7 ijms-20-01996-f007:**
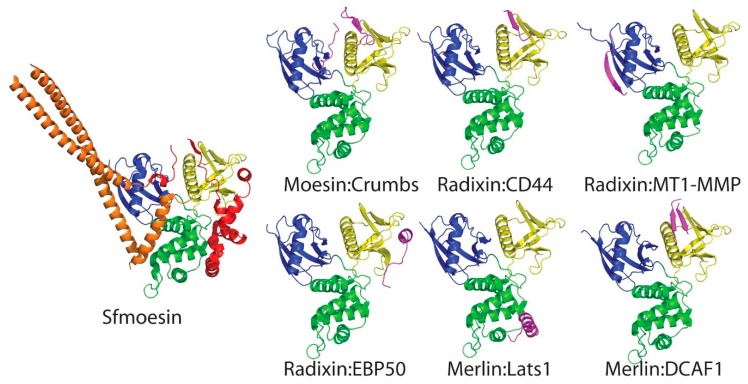
Montage of merlin-ERM proteins interacting with peptides mimicking binding partner proteins. Left panel shows the *Sf*moesin crystal structure [55] as a reference with domains colored as per Figure 1. On the right, individual panels show FERM:peptide complexes with the FERM colored as per Figure 1 and the peptides shown in magenta. The complexes are: moesin:Crumbs (PDB accession 4YL8 [107]); radixin:CD44 (2ZPY [104]); radixin:MT1-MMP (3X23 [106]); radixin:EBP50 (2D10 [108]); merlin:Lats1 (4ZRK [62]).; and merlin:DCAF1 (4P7I [111]). We note that for all complexes, with the exception of the radixin:MT1-MMP complex, the structure of the bound peptide mimics a portion of the CTD domain in the *Sf*moesin structure (compare individual complex panels with the *Sf*moesin structure). All structures are oriented the same way for direct comparison. The panels were rendered in PyMOL [63].

**Figure 8 ijms-20-01996-f008:**
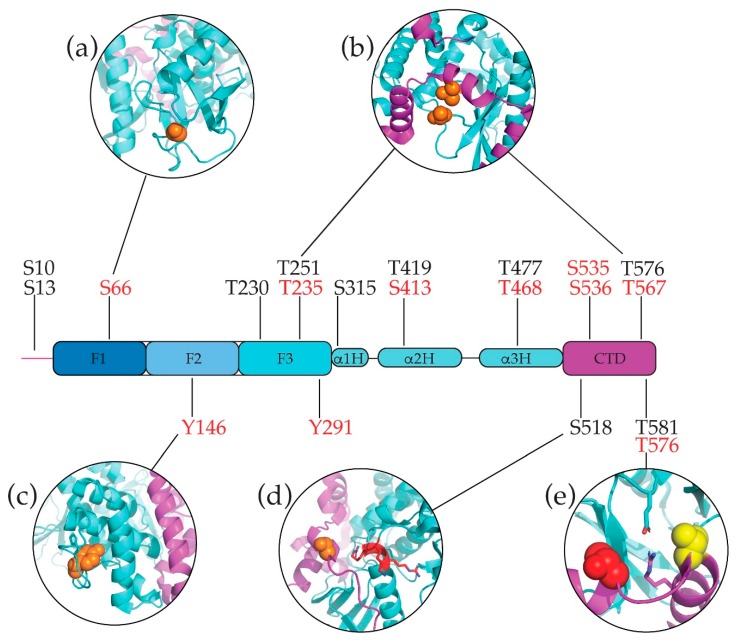
Structural landscape of some of merlin-ERM’s phosphorylation sites which are represented as atom spheres. (**a**) ezrin Ser66, a threonine in moesin and radixin and not conserved in merlin. This residue is located in subdomain F1 in the long loop between β strands β4F1 and β5F1. (**b**) Two conserved threonines, one of which (merlin Thr251, ezrin Thr235; lower one in panel) lies in a conserved loop between β strands β3F3 and β4F3 in subdomain F3, while the other is located on α3C (merlin Thr576, ezrin Thr567; upper one in panel). They are directly opposite each other and phosphorylation of both would likely destabilize the FERM–CTD interaction. (**c**) ERM Tyr146 that faces into the center of subdomain F2. In merlin and *Sf*moesin this residue is a phenylalanine, preserving the aromaticity. (**d**) When mapped onto the full-length *Sf*moesin structure, merlin Ser518 lies directly opposite one of the most conserved regions in all merlin-ERM proteins: ^308^MRRRK^312^ at the C-terminus of α1F3 in subdomain F3 (shown in red) which is highly positively charged. (**e**) Conserved ERM threonine (ezrin Thr576 but Ala585 in merlin; yellow) and merlin Thr581 (but conserved ERM arginine, ezrin Arg572, red). The two phosphorylatable threonines are proximal to the highly conserved and functionally important salt bridge (merlin Glu260–Arg588; ezrin Glu244–Arg579; Section 4.4). The panels were rendered in PyMOL [63].

**Figure 9 ijms-20-01996-f009:**
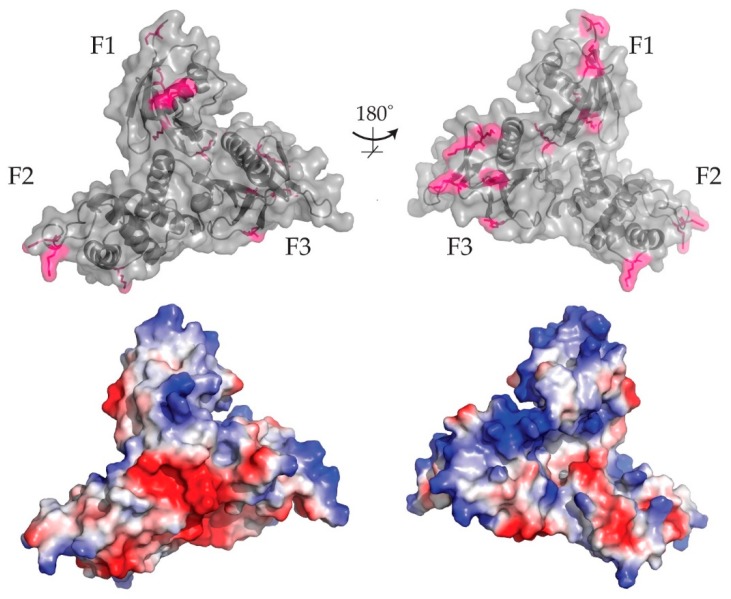
Ubiquitination and acetylation sites (pink) mapped onto the crystal structure of ezrin FERM domain (surface rendered in grey), shown in two orientations related by 180° rotation about a vertical axis in the plane of the page (top two panels). The two lower panels show the local electrostatic potential surfaces for the same two orientations of ezrin FERM when it is unmodified (i.e., not ubiquitinated nor acetylated). The coloring scheme is red for negative electrostatic potential through to blue for positive electrostatic potential. Image rendered in PyMOL [63] showing Poisson–Boltzmann electrostatic potential calculated with PDB2PQR [66].

**Figure 10 ijms-20-01996-f010:**
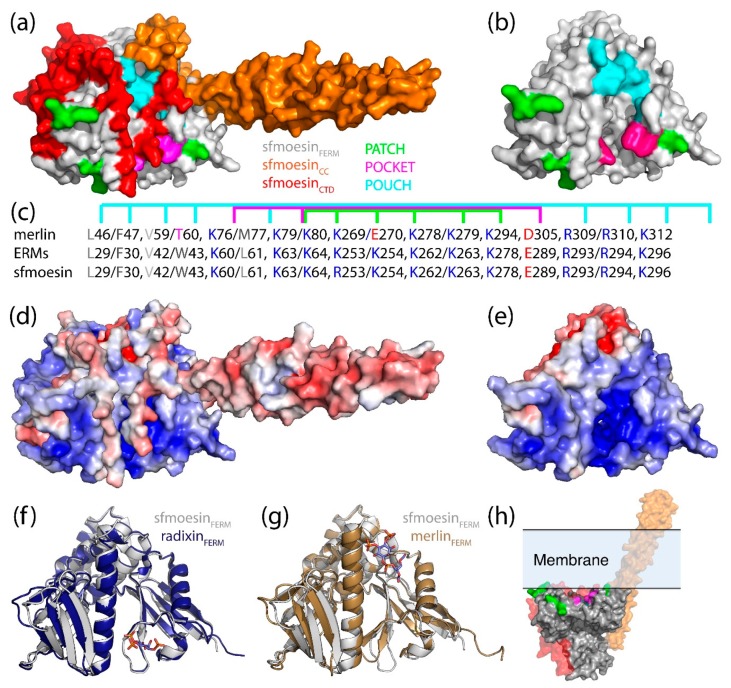
Structural insights into merlin-ERM lipid binding. (**a**) Structure of the full-length *S*f*m*oesin (PDB: 2I1K) indicating the three proposed lipid binding sites; the Patch (green), the Pocket (magenta) and the Pouch (cyan). The FERM domain is shown in grey, with the helical domain orange and the CTD in red. (**b**) The lipid-binding sites on the *Sf*moesin FERM domain in the same orientation and coloring as per (**a**). (**c**) Sequence conservation of the lipid-binding sites. Coloring for the different lipid-binding sites are consistent across panels (**a**–**c**). (**d**) and (**e**) Electrostatic maps of the full-length *Sf*moesin and the *Sf*moesin FERM domain, respectively. The coloring scheme is red for negative electrostatic potential through to blue for positive electrostatic potential. The orientation of these structures is consistent with panels (**a**) and (**b**). (**f**) Structural overlay of the *Sf*moesin FERM domain (grey) with the radixin FERM (blue) bound to IP3 (sticks representation) in the Pocket (PDB: 1GC6). (**g**) Structural overlay of the *Sf*moesin FERM (grey) with merlin FERM (gold) bound to a short chain PI(4,5)P_2_ (sticks representation) in the Pouch (PDB: 6CDS). The overlays in Panels (**f**) and (**g**) are in the same orientation as *Sf*moesin FERM in Panel (**b**). (**h**) Orientation of the full-length *Sf*moesin monomer structure docked onto a lipid bilayer. Colors are as per Panel (**a**). One can see segments of the CTD that lie between the putative lipid binding site and the membrane (red). Additionally, the location of the helical domain (orange) is incompatible with the docking of *Sf*moesin to the lipid bilayer. Image rendered in PyMOL [63] showing Poisson–Boltzmann electrostatic potential calculated with PDB2PQR [66].

**Table 1 ijms-20-01996-t001:** Sequence identity for multiple alignments between the human merlin-ERM paralogs dissected into domains. The Table entries give the percent sequence identity. In parentheses beneath each percentage, the number of identical residues is given divided by the length of the alignment, which was carried out using ClustalW [20].

	Merlin-ERM	ERM only
**Full-length**	37% (226/606)	67% (394/586)
**FERM domain** (Merlin: 1–312; ERM: 1–296)	54% (169/313)	81% (240/296)
**Helical domain** (Merlin: 313–478; ERM: 297–469)	23% (39/173)	56% (99/173)
**C-terminal domain (including polyPro tract)** (Merlin: 479–595; ERM: 470-C-terminus)	15% (15/126)	48% (57/118)

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
