# Peer review of "Two Sides of the Coin: Ezrin/Radixin/Moesin and Merlin Control Membrane Structure and Contact Inhibition"

_ijms, 2019, doi:10.3390/ijms20081996_

Round 1
Reviewer 1 Report
This is a well-drafted review article that nicely summarized the current research progress for the ERM protein family with a particular focus on the tumor suppressor member merlin/NF2. The authors systematically reviewed this key protein family from multiple perspectives including evolution, protein domain structures, post-translational modifications and several functionally associated molecular complex with proteins, lipids and cytoskeletons. The provided figures and tables are concise and informative. The cited references are accurate and sufficient. Acceptance in current format is suggested.
Author Response
We have checked the manuscript for spelling and typographical errors and we have corrected these.
Reviewer 2 Report
This manuscript is a thorough and detailed review of the structures and function of the merlin-ezrin/radixin/moesin family of proteins. Overall, I found the manuscript to be well written and filled with valuable information. My principal criticism is that there were many cases where citations were lacking. There are also some small typos throughout. My specific criticisms are listed below:
1. The overview section seems to be almost completely lacking in citations. Even for more general information citations are very useful for the interested reader, as they allow the reader to take a deeper look at topics of interest.
2. Line 56 on page 2 seems to me to be a bit of an oversimplification. While the primary role of ERMs is to link the membrane to the cytoskeleton, they also serve an important role in recruiting proteins to the membrane, as you describe later in the text. I feel it would be better to rephrase this and emphasize the difference in localization, i.e., merlin can function in the nucleus, while ERMs are mostly restricted to the membrane.
3. The beginning of section 3 also seems to be lacking in citations. The sources of the information regarding merlin-ERM genes in varying organisms should be cited.
4. Figures should have citations to the sources of the data represented in the figures.
5. In table 1, it isn’t immediately clear what the numbers are referring underneath the percentages. A brief description in the table caption would help to eliminate any potential confusion.
6. In section 4, it would be nice if the experimental details used to solve the protein structure were briefly described, since this structure is of such importance to the review.
7. Lines 514 and 515 on page 15 seem to be very similar. I would suggest combining these two lines and rephrasing.
8. The cellular data referred to in line 519 does not include a citation. Is this the same paper as reference 83? Please clarify.
9. In the beginning of section 5, it would be helpful if the authors’ indicated what protein interactions have been shown in vivo vs. in vitro. This could be helpful in other areas of the text as well where it is not immediately apparent whether something was in vivo or in vitro.
10. Line 616 has a typo: ‘plamsa‘ should be ‘plasma’
11. Line 658 has a typo: ‘immedaitely’ should be ‘immediately’
12. Line 664 seems a bit strange considering high throughput proteomic studies were already performed as stated earlier in the text. Of course there is always the possibility that something could have been missed, but I feel like this line is not really necessary unless there was some limitation to the previous work that should be addressed. In which case, the authors’ should briefly mention this limitation.
13. The paragraphs starting on lines 682 and 686 should be combined into a single cohesive unit, as they both discuss the same topic
14. Line 694 has a typo: ‘reside’ should be ‘residue’
15. I don’t feel the “(see below)” on line 697 is necessary. The discussion of the conflict begins immediately in the following paragraph.
16. Section 6 also seems to have some paragraphs that are lacking in citations, particularly 6.1.3. As a reader, I would prefer extra, potentially duplicate citations, rather than excluding them, as it makes it immediately clear where the reader should look for more information if they so desire.
17. Line 930 has a typo: ‘protein’ should be ‘proteins’
18. Line 973 has a typo: ‘bind membrane’ should be ‘bind the membrane’
19. In the paragraph starting on line 996 (page 27), PI(4,5)P2 is referenced several times in several different studies. It is not clear what form of PI(4,5)P2 is referenced here. Where these all short chain PI(4,5)P2, as mentioned in the text above? Or where some of these performed with natural long chain PI(4,5)P2?
20. Line 1006 has a typo: ‘conformational that’ should be ‘conformational change that’
Author Response
We have addressed all points raised by the reviewer.
Reviewer 2:
This manuscript is a thorough and detailed review of the structures and
function of the merlin-ezrin/radixin/moesin family of proteins. Overall, I found
the manuscript to be well written and filled with valuable information. My
principal criticism is that there were many cases where citations were lacking.
There are also some small typos throughout. My specific criticisms are listed
below:
1. The overview section seems to be almost completely lacking in citations.
Even for more general information citations are very useful for the interested
reader, as they allow the reader to take a deeper look at topics of interest.
Done. We have added references to the Overview section (Lines 42-70)
2. Line 56 on page 2 seems to me to be a bit of an oversimplification. While
the primary role of ERMs is to link the membrane to the cytoskeleton, they also
serve an important role in recruiting proteins to the membrane, as you describe
later in the text. I feel it would be better to rephrase this and emphasize the
difference in localization, i.e., merlin can function in the nucleus, while ERMs
are mostly restricted to the membrane.
Done. The initial sentences contrasting merlin function to the ERMs have been altered, referencing the distinction in cell localization and broadening the function of ERMs (lines 58-60).
3. The beginning of section 3 also seems to be lacking in citations. The
sources of the information regarding merlin-ERM genes in varying organisms
should be cited.
Much of the material in Section 3.1 is original, based on our own, previously unpublished sequence searches, multiple sequence alignments and generation of phylogenetic trees. To explain this, we have added a paragraph at the top of this section (Lines 86-90). We have also referenced the work of Golovnina et al., 2005 on the evolution of the merlin-ERM family (lines 91 and 96).
4. Figures should have citations to the sources of the data represented in the
figures.
Done. We have added citations in all Figure captions giving the sources of the data and programs used to generate the Figures.
5. In table 1, it isn’t immediately clear what the numbers are referring
underneath the percentages. A brief description in the table caption would help
to eliminate any potential confusion.
Done. We have added two sentences explaining the numbers underneath each percentage in Table 1. (Lines 134-136)
6. In section 4, it would be nice if the experimental details used to solve the
protein structure were briefly described, since this structure is of such
importance to the review.
Done. We have added experimental details used to solve the structures in paragraphs 2 and 3 of Section 4 (lines 255-257, 268-270 and 273-277)
7. Lines 514 and 515 on page 15 seem to be very similar. I would suggest
combining these two lines and rephrasing.
We have reversed the order of these two sentences and elaborated on the new opening sentence so as to clarify the meaning. The new second sentence now follows from the first (lines 554-556)
8. The cellular data referred to in line 519 does not include a citation. Is this
the same paper as reference 83? Please clarify.
Done. The reference is indeed 83. We have added the explicit reference to clarify this (line 559).
9. In the beginning of section 5, it would be helpful if the authors’ indicated
what protein interactions have been shown in vivo vs. in vitro. This could be
helpful in other areas of the text as well where it is not immediately apparent
whether something was in vivo or in vitro.
This is beyond the scope of this review. We have added three sentences at the end of the first paragraph of Section 5 explaining this and referring the reader to the primary references cited in the text (line 608-611).
10. Line 616 has a typo: ‘plamsa‘ should be ‘plasma’
Done (line 660)
11. Line 658 has a typo: ‘immedaitely’ should be ‘immediately’
Done (line 705)
12. Line 664 seems a bit strange considering high throughput proteomic
studies were already performed as stated earlier in the text. Of course there is
always the possibility that something could have been missed, but I feel like
this line is not really necessary unless there was some limitation to the
previous work that should be addressed. In which case, the authors’ should
briefly mention this limitation.
This sentence has now been removed (line 710).
13. The paragraphs starting on lines 682 and 686 should be combined into a
single cohesive unit, as they both discuss the same topic
Done. Paragraphs joined into one unit (line 728-735).
14. Line 694 has a typo: ‘reside’ should be ‘residue’
Done (line 740)
15. I don’t feel the “(see below)” on line 697 is necessary. The discussion of
the conflict begins immediately in the following paragraph.
Removed “(see below)” (line 743)
16. Section 6 also seems to have some paragraphs that are lacking in
citations, particularly 6.1.3. As a reader, I would prefer extra, potentially
duplicate citations, rather than excluding them, as it makes it immediately clear
where the reader should look for more information if they so desire.
Done. The residues discussed in Section 6.1.3 were identified on PhosphoSitePlus, as discussed in the opening paragraph to section 6. We have added explicit references to this throughout Section 6.1.3 (lines: 801, 808, 819 and 830).
17. Line 930 has a typo: ‘protein’ should be ‘proteins’
Done (line 981)
18. Line 973 has a typo: ‘bind membrane’ should be ‘bind the membrane’
Done (line 1024)
19. In the paragraph starting on line 996 (page 27), PI(4,5)P2 is referenced
several times in several different studies. It is not clear what form of PI(4,5)P2
is referenced here. Where these all short chain PI(4,5)P2, as mentioned in the
text above? Or where some of these performed with natural long chain
PI(4,5)P2?
Done. We have clarified all instances where non-natural forms of PI(4,5)P2 are used and added a statement indicating this (lines 1047-1067).
20. Line 1006 has a typo: ‘conformational that’ should be ‘conformational
change that’
Done (Line 1059)